# Distributionally Robust Policy Evaluation under General Covariate Shift in Contextual Bandits

**Yihong Guo**[*]                                                                                *yguo80@jhu.edu*
*Department of Computer Science, Johns Hopkins University*
**Hao Liu**[*]                                                                                      *hliu3@caltech.edu*
*Department of Computing and Mathematical Sciences, Caltech*
**Yisong Yue**                                                                                    *yyue@caltech.edu*
*Department of Computing and Mathematical Sciences, Caltech*
**Anqi Liu**                                                                                        *aliu@cs.jhu.edu*
*Department of Computer Science, Johns Hopkins University*

**Reviewed on OpenReview:** *https://openreview.net/forum?id=R7PReNELww*

## Abstract

We introduce a distributionally robust approach that enhances the reliability of offline policy evaluation in contextual bandits under general covariate shifts. Our method aims to deliver robust policy evaluation results in the presence of discrepancies in both context and policy distribution between logging and target data. Central to our methodology is the application of robust regression — a distributionally robust technique tailored here to improve the estimation of conditional reward distribution from logging data. Utilizing the reward model obtained from robust regression, we develop a comprehensive suite of policy value estimators, by integrating our reward model into established evaluation frameworks, namely direct methods and doubly robust methods. Through theoretical analysis, we further establish that the proposed policy value estimators offer a finite sample upper bound for the bias, providing a clear advantage over traditional methods, especially when the shift is large. Finally, we designed an extensive range of policy evaluation scenarios, covering diverse magnitudes of shifts and a spectrum of logging and target policies. Our empirical results indicate that our approach significantly outperforms baseline methods, most notably in 90% of the cases under the policy shift-only settings and 72% of the scenarios under the general covariate shift settings.

## 1 INTRODUCTION

Contextual bandits are online learning algorithms where a policy learner repeatedly observes a context, takes an action, and receives a reward only for the selected action (Langford & Zhang, 2007). It has a wide range of real-world applications, including recommender systems (Li et al., 2010; Yue et al., 2012; Joachims et al., 2021), online advertising (Tang et al., 2013; Bottou et al., 2013), experimental design (Krause & Ong, 2011), and medical interventions (Lei et al., 2014; Tewari & Murphy, 2017). Evaluating policy performance is essential for these applications, but assessing the value through direct online deployment can be costly and impractical. This motivates the important task of policy evaluation from historical, pre-collected data.

The key challenge in policy evaluation using pre-collected data lies in addressing the counterfactual reasoning that arises from the divergence between the distributions of the logging and target data. Specifically, the actions selected by the target policy may not be covered in the pre-collected logging data, and shifts in the context distribution can further complicate policy evaluation. When the distribution shift only occurs in the policy distribution, we recover the classic setting of off-policy evaluation (OPE). Existing OPE approaches

---

[*]These authors contributed equally to this work and are listed in alphabetical order.

can be divided into three main categories: (1) inverse propensity scoring (IPS), which uses importance weights adjustment (Horvitz & Thompson, 1952; Swaminathan & Joachims, 2015b); (2) direct methods (DM), which directly regress the value of a target policy by learning a conditional reward model; and (3) doubly robust (DR) methods, which integrate DM and IPS (Bang & Robins, 2005; Dudik et al., 2011; Wang et al., 2017; Su et al., 2019a; Dudik et al., 2014), achieving asymptotic consistency if either DM or IPS is consistent. In both DM and DR methods, a high-quality reward estimation for each context-action pair in the target data can be beneficial. However, compared to the rich literature on improving the IPS component (Swaminathan & Joachims, 2015b; Wang et al., 2017; Su et al., 2019a), less work has explored how to integrate a carefully designed conditional reward model in OPE (Farajtabar et al., 2018).

Beyond the standard setting of policy shift discussed above, very few studies have explored distribution shift in the context variables. Uehara et al. (2020) suggested incorporating the context distribution into IPS estimation for both IPS and DR methods when a context shift occurs and may suffer from similar issues like high variance as traditional OPE methods. However, their approach does not address how a shift in context distribution might influence reward estimation. Recently, distributionally robust methods have been introduced to enhance the robustness of the evaluation by constructing KL-divergence uncertainty sets to capture potential distribution shifts (Si et al., 2020; Kallus et al., 2022) in the joint distribution of context, action, and reward. However, the substantially large uncertainty set that characterizes the shift may lead to over-conservativeness in practice.

In this paper, we tackle the problem of policy evaluation through the lens of covariate shift (Shimodaira, 2000). Covariate shift refers to the modeling of a dependent variable when the marginal distribution of the covariates differs between training and test time but the conditional output distribution on the covariates stays the same. Policy evaluation can be interpreted as learning under the covariate shift, where the dependent variable is the reward, the covariates are the contexts and actions, and we are interested in learning a robust conditional reward model $\hat{r}(\boldsymbol{x}, a) \sim P(r|\boldsymbol{x}, a)$ that performs well under distribution shift. Specifically, we study the settings where shifts can occur in policy distribution alone (the standard setting), and both policy and context distributions, which we refer to as *Policy Shift* (PS) and *General Covariate Shift* (GCS), respectively. Building on recent progress in robust regression under covariate shift (Chen et al., 2016; Liu et al., 2019), we introduce a novel family of distributionally robust policy evaluation estimators, which is created by seamlessly integrating the conditional reward model, obtained from robust regression, into existing DM and DR methods. Our contributions are as follows:

- We formulate the task of policy evaluation using the framework of covariate shift and incorporate a robust regression method into the policy evaluation setting where shifts can happen in both policy and context distribution. Leveraging a base distribution, the resulting shift-aware conditional reward model allows for a conservative evaluation of policy value in scenarios where the context-action pair in target data is not encompassed by the pre-collected logging data.

- We develop direct methods based on the proposed shift-aware conditional reward model, which can be easily integrated into various doubly robust methods. We establish the finite sample upper bound results for the bias of the proposed method, demonstrating the proposed method's robustness under large distribution shifts, which is further validated in the experiments.

- We conduct extensive experiments spanning 360 policy shift conditions and 1260 total conditions for various combinations of logging, target policies, and covariate shifts on data. In our comparison against strong baseline methods on standard benchmark datasets, our method demonstrates superior performance under distribution shifts across various conditions, outperforming the baseline methods in over 90% of cases under policy shifts, and over 72% of the instances in general covariate shift settings.

## 2 Related Work

**Off-Policy Evaluation for Contextual Bandits.** Classic methods for OPE include direct methods (DM), inverse propensity scoring (IPS), and doubly robust (DR) methods that combine DM and IPS. DM can suffer from significant bias Dudik et al. (2011). IPS, though unbiased, exhibits high variance. DR methods (Dudik et al., 2011; 2014; Wang et al., 2017; Farajtabar et al., 2018) balance the strengths of both. If either IPS

or DM is unbiased, DR methods are guaranteed to be unbiased (Dudik et al., 2011). Much of the follow-up work on DR methods has focused on mitigating the detrimental effects of variance or extreme probabilities from the IPS component. For example, SnDR utilizes SnIPS (Swaminathan & Joachims, 2015b), SWITCH estimator truncates IPS using a threshold (Wang et al., 2017), and DR with Optimistic Shrinkage (DRoS) finds a weighting strategy by optimizing a sharp bound on mean squared error (Su et al., 2019a). While much of the prior work has emphasized improving the IPS component of DR estimators and their asymptotical guarantees, our research targets enhancing the robustness of conditional reward estimators, especially under large shifts and limited data. Our key insight is to view this problem as a covariate shift problem and utilize robust regression to train the conditional reward estimator.

**Covariate Shift and Distributional Robustness.** Covariate shift has been tackled through methods such as importance weighting (Shimodaira, 2000; Sugiyama et al., 2007) and kernel mean matching (Gretton et al., 2009) in machine learning. In policy evaluation area, Uehara et al. (2020) introduced a DR-based estimator under covariate shift that uses a density ratio estimator between the source and target distributions. Their method focuses on the propensity score weighting formulation, and the IPS component in the estimator introduces high variance when extreme weights exist. In contrast, we tackle the problem orthogonally, utilizing a distributionally robust direct method that can be effortlessly combined with their method.

There are several studies exploring distributionally robust optimization (DRO) (Ben-Tal et al., 2009; Słowik & Bottou, 2022; Blanchet et al., 2021) in policy evaluation. Si et al. (2020) and Kallus et al. (2022) augmented evaluation robustness by conservatively estimating reward mapping with KL-divergence uncertainty sets. However, in these methods, the uncertainty set covers all possible shifts in the joint distribution of context, action, and reward, which does not characterize the covariate shift specifically. As a result, the estimator can be overly robust and less effective. Another line of work utilize DRO for estimating confidence interval in the off-policy setting (Karampatziakis et al., 2020; Dai et al., 2020; Faury et al., 2020), following a generalized empirical likelihood approach (Duchi et al., 2021). The differences from our approach lie in the formulation of the DRO problem. Specifically, they do not use the conditional distribution of the target variable (the reward in our case) as the adversarial player, and the uncertainty set constraint is formulated by the perturbation of empirical distribution of the data with distance measured using f-divergence, rather than using feature matching as in our case. Robust regression (Chen et al., 2016; Liu & Ziebart, 2017) tailored distributional robustness methods specifically for the setting of covariate shift. It constructs a predictor that approximately matches training data statistics but is otherwise the most uncertain on the testing distribution by minimizing the worst-case expected target loss and obtaining a parametric form of the predicted output labels' probability distributions. We are the first to employ robust regression methods for policy evaluation, enabling the refinement of direct methods and the further enhancement of doubly robust estimators.

**Detailed Related Work.** We provide a more detailed review of related work in Appendix A.

## 3 Problem Description

**Policy Evaluation for Contextual Bandits.** In contextual bandits, a policy $\pi$ iteratively observes a context $\boldsymbol{x}$, takes an action $a$, and receives a scalar reward $r_{\boldsymbol{x},a}$. Assuming that the contexts $\boldsymbol{x}$ are independent and identically distributed (i.i.d.), the target object, the value of the target policy, can be expressed as:

$$V^T = \mathbb{E}_{\boldsymbol{x} \sim P_t(\boldsymbol{x}), a \sim \pi(a|\boldsymbol{x})} \left[ r_{\boldsymbol{x},a} \right], \tag{1}$$

where $\boldsymbol{x} \sim P_t(\boldsymbol{x})$ denotes some known exogenous target context distribution (e.g., profiles of users), and $a \sim \pi(a|\boldsymbol{x})$ denotes the stochasticity of the known target policy. Ususally, we assume access to a pre-collected dataset of $n$ tuples of the form: $S = \{(\boldsymbol{x}, a, r)\}$, where $\boldsymbol{x} \sim P_s(\boldsymbol{x})$, $a \sim \beta(a|\boldsymbol{x})$, and $r$ is the reward observed. Then given $S$ and $\pi$, the goal of policy evaluation is to reliably estimate $\hat{V}_S^T \equiv \hat{V}$ of $V^T$ in Equation 1. In traditional OPE, the goal is to estimate $V^T$ offline using pre-collected historical data from some other (possibly unknown) logging policy $\beta(a|\boldsymbol{x})$ on the context distribution $\boldsymbol{x} \sim P_s(\boldsymbol{x})$, with the assumption that $P_s(\boldsymbol{x}) = P_t(\boldsymbol{x})$. When we consider context distribution shift, we further assume $P_s(\boldsymbol{x}) \neq P_t(\boldsymbol{x})$. Here, $s$ and $t$ stand for source and target distribution, respectively.

**Policy Evaluation as Covariate Shift.**

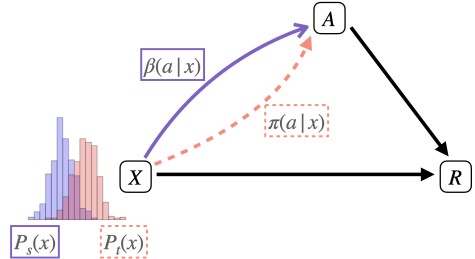

Covariate shift is a special case of distribution shift where only the distribution of the covariates changes, but the conditional output distribution remains unaltered (Shimodaira, 2000). In the setting of this paper, our goal is to evaluate policy $\pi$ over a potentially different context distribution $P_t(\boldsymbol{x})$, as depicted in the data-generating process in Figure 1. The assumption can be expressed as $P_s(\boldsymbol{x}, a) \neq P_t(\boldsymbol{x}, a)$ and $P_s(r|\boldsymbol{x}, a) = P_t(r|\boldsymbol{x}, a)$. Note we use "logging" and "source" distribution interchangeably. Based on this data-generating process, the joint distribution of covariates ($\boldsymbol{x}$ and $a$ in this case) can be described as shifting from $P_s(\boldsymbol{x}, a) = P_s(\boldsymbol{x})\beta(a|\boldsymbol{x})$ to $P_t(\boldsymbol{x}, a) = P_t(\boldsymbol{x})\pi(a|\boldsymbol{x})$. We can further assume that the distribution shift problem of policy evaluation can be decomposed into two different mechanisms (Schölkopf et al., 2012; Peters et al., 2017): (1) Context distribution shift, and (2) Policy shift. General covariate shift (GCS) refers to the setting when both context distribution shift and policy shift occur. As the ground truth conditional reward distribution $P^*(r|\boldsymbol{x}, a)$, that we aim to learn, remains unchanged, we can formulate the reward estimation problem in policy evaluation as a covariate shift problem. To simplify the notations, We omit $\boldsymbol{x} \sim P(\boldsymbol{x})$ sometimes in the OPE setting. We further define $P_s(\boldsymbol{x}, a, r) = P_s(\boldsymbol{x}, a)P^*(r|\boldsymbol{x}, a)$ and $P_t(\boldsymbol{x}, a, r) = P_t(\boldsymbol{x}, a)P^*(r|\boldsymbol{x}, a)$.

Figure 1: Data generating process for policy evaluation under general covariate shift. $X$, $A$, and $R$ represent context, action, and reward variables, respectively. Logging data distribution $P_s(x)\beta(a|\boldsymbol{x})$ is marked with solid purple lines, and target data distribution $P_t(x)\pi(a|\boldsymbol{x})$ is marked with dashed red lines. Black lines indicate reward is generated by both contexts and actions.

## 4 Distributionally Robust Policy Evaluation under General Covariate Shift

In this section, we formally present our robust regression approach for reward prediction from logging data in policy evaluation, which is constructed from a distributionally robust learning framework. The high-level goal is to estimate a reward model $\hat{f}(\boldsymbol{x}, a)$ that can be robust to the worst-case data-generating distribution that can occur to maximize a loss function on the target data. We present the formulation and derived predictive form in Section 4.1 and the parameter learning algorithm in Section 4.2.

### 4.1 The Distributionally Robust Learning Formulation

**ERM v.s. DRL.** Traditional supervised learning frames reward estimation as an empirical risk minimization problem (ERM). ERM seeks to minimize the following empirical risk on the logging data:

$$\hat{f}_{\text{sup}}(\boldsymbol{x}, a) = \arg\min \mathbb{E}_{P_s(\boldsymbol{x}, a, r)}\left[\mathcal{L}(r, \hat{f}(\boldsymbol{x}, a))\right] \approx \arg\min \frac{1}{|S|}\sum_{i \in S}\mathcal{L}(r_i, \hat{f}(\boldsymbol{x}_i, a_i)), \tag{2}$$

where $\mathcal{L}$ is a loss function that measures the disparity between the actual reward and prediction from the model. However, this approach is susceptible to bias under covariate shift, as the logging data reflect reward information solely within the scope of the logging context and policy. To address this, the Distributionally Robust Learning (DRL) framework provides an alternative perspective by modeling the problem as a two-player minimax game over the target data, which involves a predictor player minimizing and an adversarial player maximizing the expected target loss as follows: $\hat{f}(\boldsymbol{x}, a) = \arg\min_{\hat{f}(\boldsymbol{x}, a)} \max_{g(\boldsymbol{x}, a) \in \Sigma_S} \mathbb{E}_{\boldsymbol{x}, a \sim P_t(\boldsymbol{x}, a), r \sim g}\left[\mathcal{L}(r, \hat{f}(\boldsymbol{x}, a))\right]$, where $\hat{r} \sim \hat{f}(\boldsymbol{x}, a)$ is the predictor player and $r \sim g(\boldsymbol{x}, a)$ is the adversarial player. The adversarial player is confined to reside in a constrained set $\Sigma_S$, defined on the training data $S$. Covariate shift results in a mismatch between the expected loss we are optimizing (defined on $P_t(x, a)$) and the constrained set for the adversary (defined on $P_s(x, a)$). The choice of the loss function $\mathcal{L}$ and the constrained set $\Sigma_S$ shape the specific derived form of the predictor $\hat{f}(\boldsymbol{x}, a)$. Both $f$ and $g$ are valid conditional distributions of $r$ given the context and the action $(\boldsymbol{x}, a)$.

**DRL Formulation for Robust Regression.** In this paper, we incorporate the following relative loss function for the reward estimation:

$$\mathcal{L}(r, \hat{f}) := \mathbb{E}_{\boldsymbol{x}, a \sim P_t(\boldsymbol{x}, a), r \sim g}\left[-\log \hat{f} + \log f_0\right], \tag{3}$$

---

**Algorithm 1** Stochastic Gradient Descent for Robust Regression under General Covariate Shift

---

**Input**: Training data points $\{(\boldsymbol{x}_i, a_i, r_i)\}$, logging policy $\beta(a|\boldsymbol{x})$, target policy $\pi(a|\boldsymbol{x})$, source data distribution $P_s(\boldsymbol{x})$, target data distribution $P_t(\boldsymbol{x})$, $\boldsymbol{\theta}$ with initialization, SGD optimizer Opt, learning rate lr, epoch number $T$.

$\boldsymbol{\theta} \leftarrow$ random initialization, epoch $\leftarrow 0$;

**While** epoch $< T$

    **For** each mini-batch

        Compute $\mu_{\boldsymbol{\theta}}(\boldsymbol{x}, a)$ and $\sigma_{\boldsymbol{\theta}}^2(\boldsymbol{x}, a)$ following Equation 5;

        Compute gradients for $\boldsymbol{\theta}$ following Equation 6 and Equation 7;

        Stochastic Gradient descent on $\boldsymbol{\theta}$ using Opt.step(lr);

**Output**: Trained model parameters $\boldsymbol{\theta}$.

---

where $f_0$ is a predefined base predictor. This loss function represents the conditional log-loss difference between predictor $\hat{f}$ and a base conditional distribution $f_0$ with respect to $g$ on the target data distribution. Given a fixed $g$, $\hat{f}$ aims to be as close as to $g$ if possible. If this is unachievable, $\hat{f}$ will be equal to $f_0$. The motivation for introducing the base distribution $f_0$ is that the logging data is not always informative in minimizing the loss function on the target data. Therefore, there should be a default choice of prediction when logging data does not cover the target data distribution well enough. To further shape this process, we incorporate the following constrain $\Sigma_S$ (used in the definition of $\hat{f}(\boldsymbol{x}, a)$) to regulate the adversary $g$:

$$\Sigma_S := \left\{ g \left| \left| \mathbb{E}_{\boldsymbol{x}, a \sim P_s(\boldsymbol{x}, a), r \sim g}[r(\boldsymbol{x}, a)^2] - \frac{1}{|S|} \sum_{i \in S} r_i^2 \right| \le \eta, \left| \mathbb{E}_{\boldsymbol{x}, a \sim P_s(\boldsymbol{x}, a), r \sim g}[r(\boldsymbol{x}, a)\phi(\boldsymbol{x}, a)] - \frac{1}{|S|} \sum_{i \in S} r_i \phi(\boldsymbol{x}, a) \right| \le \eta \right. \right\}, \tag{4}$$

where $\phi(\boldsymbol{x}, a)$ is a feature vector and $\eta$ is the slack term. In general, $\phi(\boldsymbol{x}, a)$ can be any feature functions. In this work, we set it to be the concatenation of $\boldsymbol{x}$ and $a$. For simplicity, we use the same slack for the two constraints in Equation 4. This constraint indicates that the expectation of a quadratic function in $r$ under the adversary conditional probability distribution $g$ must be close to the empirical expectation (i.e., the average) of that function within the logging data. We choose this quadratic function because it leads to a Gaussian distribution predictor, which is an exponential family distribution with a quadratic form.

**Derived Predictive Form.** If we also choose the base distribution to be Gaussian as $f_0 \sim \mathcal{N}(\mu_0, \sigma_0^2)$, we have the following form for the predictor $\hat{f}$: $\hat{f}_{\boldsymbol{\theta}}(\boldsymbol{x}, a) \sim \mathcal{N}(\mu_{\boldsymbol{\theta}}(\boldsymbol{x}, a), \sigma_{\boldsymbol{\theta}}^2(\boldsymbol{x}, a))$ (The full derivation can be found in Appendix C):

$$\sigma_{\boldsymbol{\theta}}^2(\boldsymbol{x}, a) = \left(2\mathcal{W}(\boldsymbol{x}, a)\theta_r + \sigma_0^{-2}\right)^{-1}, \quad \mu_{\boldsymbol{\theta}}(\boldsymbol{x}, a) = \sigma_{\boldsymbol{\theta}}^2(\boldsymbol{x}, a)\left(-2\mathcal{W}(\boldsymbol{x}, a)\boldsymbol{\theta_x}\phi(\boldsymbol{x}, a) + \mu_0 \sigma_0^{-2}\right), \tag{5}$$

where $\boldsymbol{\theta_x}$ and $\theta_r$ are components in the full parameter $\boldsymbol{\theta} = [\theta_r, \boldsymbol{\theta_x}]$, and $\boldsymbol{\theta_x}$ is a vector with the same dimension as $\phi$. $\mathcal{W}(\boldsymbol{x}, a)$ is the density ratio, which equals to $\frac{P_s(\boldsymbol{x}, a)}{P_t(\boldsymbol{x}, a)}$ under general covariate shift, and reduces to $\frac{\beta(a|\boldsymbol{x})}{\pi(a|\boldsymbol{x})}$ in the OPE setting as the context distribution does not shift. The derivation involves using strong duality to switch the min and max problem and applying the Lagrangian multiplier $\boldsymbol{\theta}$ to convert the constrained problem of Equation 3 and Equation 4 into an unconstrained one.

**The Role of the Density Ratio $\mathcal{W}(x, a)$ and the Base Distribution $f_0$.** The density ratio $\mathcal{W}(\boldsymbol{x}, a)$ is defined as the ratio of the source density $P_s(\boldsymbol{x}, a)$ over the target density $P_t(\boldsymbol{x}, a)$. For cases where the magnitude of $P_s(\boldsymbol{x}, a)$ is significantly smaller than $P_t(\boldsymbol{x}, a)$, the corresponding $(\boldsymbol{x}, a)$ occurs infrequently in the logging data but frequently in the target data. Under such cases, the estimator will rely more on the base distribution $f_0$ rather than the logging data, as $\mathcal{W}(\boldsymbol{x}, a)$ becomes closer to zero in Equation 5. In other words, the estimator heavily depends on the base distribution when there is a large discrepancy between the logging and target data. On the other hand, if the logging data is well-covered, the estimator becomes more confident in its predictions, with $\mu_{\boldsymbol{\theta}}$ relying heavily on the learned parameters $\boldsymbol{\theta_x}$ and $\theta_r$. Intuitively, the base distribution will affect the result more when the distribution shift is considerably large.

**The Role of Gaussian Predictive Form.** We assume a Gaussian base distribution and a quadratic feature constraint, which results in a Gaussian conditional target distribution $p(r|\boldsymbol{x}, a)$. While this Gaussian

assumption is generally valid for real-world examples and widely applied in ML literature, we can construct synthetic examples to violate such assumption. For instance, we can define the conditional target distribution as an exponential family with cubic terms over the exponential, corresponding to a cubic function feature constraint. In such cases, the standard bias and variance tradeoff analysis applies: using a model with Gaussian assumption in this situation leads to an overly simplified model, resulting in high bias and low variance. However, in practice, when little is known about the underlying data generation process, the robust regression model under Gaussian assumption is preferred due to its nice implementation properties and broad applicability. Further discussions on the Gaussian assumption is provided in Appendix C.1.

## 4.2 The Proposed Algorithm for Parameter Learning

We summarize the proposed robust regression approach for reward prediction in Figure 2 (a). After deriving the predictive form, we here present how to learn model parameters $\boldsymbol{\theta}$. With the predictive form of $\hat{f}$ in Equation 5, we can plug it back into the Lagrangian form of the constrained optimization problem in Equation 3 and Equation 4, and obtain the target loss to be equivalent as maximizing the log-likelihood over the target distribution: $\mathcal{L}(\boldsymbol{\theta}) = \mathbb{E}_{P_t(\boldsymbol{x},a,r)}\left[\log \hat{f}_{\boldsymbol{\theta}}(\boldsymbol{x},a)\right]$. Given the form of $\hat{f}$, this objective is convex w.r.t $\boldsymbol{\theta}$. By taking the derivative of $\mathcal{L}(\boldsymbol{\theta})$ w.r.t the $\theta_r$ and $\boldsymbol{\theta_x}$, we can obtain the following gradient form, where $S_{\text{batch}}$ is the set for data mini-batch. Standard gradient-based learning can then be applied for learning the parameters. The training procedure is summarized in Algorithm 1. Detailed derivation are in Appendix C.

$$\nabla_{\theta_r}\mathcal{L}(\boldsymbol{\theta}) = \frac{1}{|S_{\text{batch}}|}\sum_{(\boldsymbol{x},a,r)\in S_{\text{batch}}}(\mu_{\boldsymbol{\theta}}^2(\boldsymbol{x},a) + \sigma_{\boldsymbol{\theta}}^2(\boldsymbol{x},a) - r^2), \tag{6}$$

$$\nabla_{\boldsymbol{\theta_x}}\mathcal{L}(\boldsymbol{\theta}) = \frac{1}{|S_{\text{batch}}|}\sum_{(\boldsymbol{x},a,r)\in S_{\text{batch}}}(\mu_{\boldsymbol{\theta}}(\boldsymbol{x},a) - r)\boldsymbol{\phi}(\boldsymbol{x},a). \tag{7}$$

# 5 Methods

In this section, we first provide an overview of how our robust regression approach can be integrated into existing policy evaluation methods as a reward model. A detailed comparison between classical OPE and the setting of policy evaluation under general covariate shift considered in this paper is illustrated in Figure 2 (b). We then establish upper bounds for the bias of these methods. We will refer to our methods using the notation **PS** and **GCS** as suffixes, representing policy shift and general covariate shift, respectively. Specifically, the proposed methods **DM-PS**, **DR-PS**, **DM-GCS**, **DR-GCS**, **SnDR-PS** and **SnDR-GCS** are formulated as follows. We also investigate the integration of our robust regression approach into other OPE methods including SWITCH (Wang et al., 2017) and DRoS (Su et al., 2020) in Appendix E.4.

**DM-PS**. In the OPE setting, the context distribution $P(\boldsymbol{x})$ does not shift, reducing the density ratio $\mathcal{W}(\boldsymbol{x},a)$ to $\frac{\beta(a|\boldsymbol{x})}{\pi(a|\boldsymbol{x})}$. We enhance standard DMs by incorporating the mean value predictor estimated from robust regression, denoted as $\mu_{\boldsymbol{\theta}}(\boldsymbol{x},a)$, yielding the "Direct Method with Policy Shift" (DM-PS) estimator.

$$\hat{V}_{\text{DM-PS}} = \frac{1}{|S|}\sum_{\boldsymbol{x}\in S}\mathbb{E}_{a\sim\pi}\left[\mu_{\boldsymbol{\theta}}^{\text{PS}}(\boldsymbol{x},a)\right]. \tag{8}$$

**DR-PS**. We apply DM-PS to be the reward model part in DR framework, yielding the "Doubly Robust estimator with Policy Shift" (DR-PS) estimator.

$$\hat{V}_{\text{DR-PS}} = \frac{1}{|S|}\sum_{(\boldsymbol{x},a,r)\in S}\left[\frac{(r-\mu_{\boldsymbol{\theta}}^{\text{PS}}(\boldsymbol{x},a))\pi(a|\boldsymbol{x})}{\hat{\beta}(a|\boldsymbol{x})}\right] + \hat{V}_{\text{DM-PS}}. \tag{9}$$

**DM-GCS**. Under general covariate shift, we set $\mathcal{W}(\boldsymbol{x},a)$ to be $\frac{P_s(\boldsymbol{x},a)}{P_t(\boldsymbol{x},a)}$ in robust regression and plug it into the standard DM to get "Direct Method with General Covariate Shift" (DM-GCS) estimator. We also plug $\mathcal{W}(\boldsymbol{x},a) = \frac{P_s(\boldsymbol{x},a)}{P_t(\boldsymbol{x},a)}$ into IPS to implement IPS-GCS as a baseline. More details can be found in Appendix B.

$$\hat{V}_{\text{DM-GCS}} = \frac{1}{|S|}\sum_{\boldsymbol{x}\in S}\mathbb{E}_{a\sim\pi}\left[\mu_{\boldsymbol{\theta}}^{\text{GCS}}(\boldsymbol{x},a)\right], \tag{10}$$

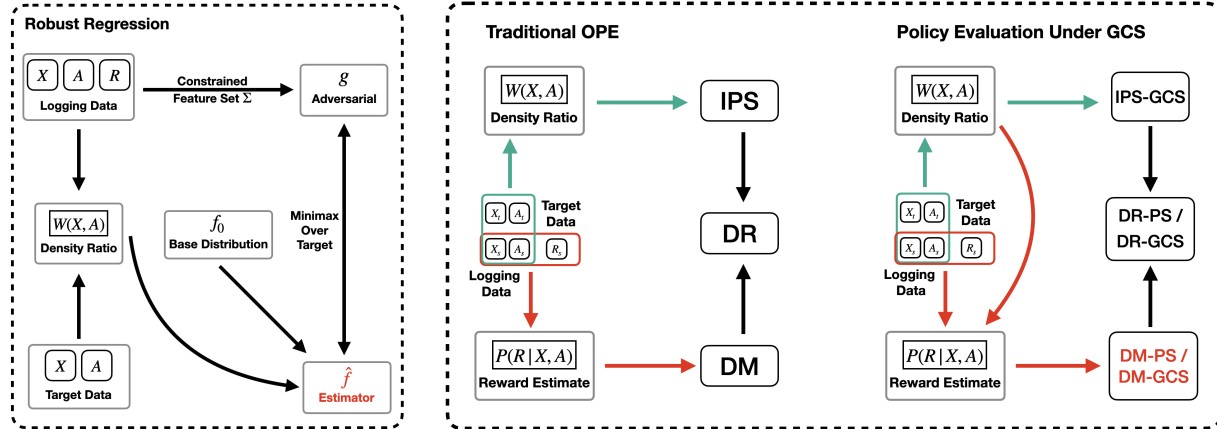

(a) The Robust Regression Method      (b) Comparison between OPE and the proposed approach

Figure 2: **(a):** Description of the robust regression method. The density ratio $\mathcal{W}(\boldsymbol{x}, a)$ is estimated from the logging data and the target data. The feature set $\Sigma$ is obtained from logging data to constrain the adversarial player $g$. The estimator $\hat{f}$ is estimated by taking minimax loss over target data distribution, with density ratio $\mathcal{W}(\boldsymbol{x}, a)$, base distribution $f_0$, and adversarial player $g$. **(b):** Comparision between the setting of classical off-policy evaluation (OPE) and policy evaluation under general covariate shift. There are two types of information derived from the data: (1) density ratio information, obtained from the covariate (context and action) distribution from both logging and target data and used by the inverse propensity score (IPS) method (green arrow), and (2) pairing information between covariate and reward from the logging data, used by the direct method (DM) (red arrow) to estimate the reward conditional distribution. Traditional DM relies solely on the logging data, whereas the proposed DM-PS and DM-GCS methods also utilize the density ratio information in the robust regression.

**DR-GCS**. We apply DM-GCS to be the reward model part in DR and apply IPS-GCS to the IPS part of DR to obtain the "Doubly Robust estimator with General Covariate Shift" (DR-GCS) estimator.

$$\hat{V}_{\text{DR-GCS}} = \frac{1}{|S|} \sum_{(\boldsymbol{x},a,r) \in S} \left[ \frac{(r - \mu_{\boldsymbol{\theta}}^{\text{GCS}}(\boldsymbol{x},a)) P_t(\boldsymbol{x},a)}{P_s(\boldsymbol{x},a)} \right] + \hat{V}_{\text{DM-GCS}}. \tag{11}$$

**SnDR-PS/SnDR-GCS**. Similar to standard DR, following the idea of SnIPS, we can further control the variance of the IPS component and obtain the self-normalized DR-PS (SnDR-PS) and self-normalized DR-GCS (SnDR-GCS), by normalizing over the sum of the importance weights.

$$\hat{V}_{\text{SnDR-PS}} = \frac{1}{|S|} \sum_{(\boldsymbol{x},a,r) \in S} \left[ \frac{(r - \mu_{\boldsymbol{\theta}}^{\text{PS}}(\boldsymbol{x},a)) \pi(a|\boldsymbol{x})}{\hat{\beta}(a|\boldsymbol{x}) \sum_{(\boldsymbol{x},a,r) \in S} \frac{\pi(a|\boldsymbol{x})}{\hat{\beta}(a|\boldsymbol{x})}} \right] + \hat{V}_{\text{DM-PS}}, \tag{12}$$

$$\hat{V}_{\text{SnDR-GCS}} = \frac{1}{|S|} \sum_{(\boldsymbol{x},a,r) \in S} \left[ \frac{(r - \mu_{\boldsymbol{\theta}}^{\text{GCS}}(\boldsymbol{x},a)) P_t(\boldsymbol{x},a)}{P_s(\boldsymbol{x},a) \sum_{(\boldsymbol{x},a,r) \in S} \frac{P_t(\boldsymbol{x},a)}{P_s(\boldsymbol{x},a)}} \right] + \hat{V}_{\text{DM-GCS}}. \tag{13}$$

## 5.1 Theoretical Results

### 5.1.1 Bias Analysis

**Expected Error.** We first establish the upper bound for the robust regression-based reward estimation's expected squared error on the target distribution as follows:

**Theorem 1.** *Assuming the density ratio of the target over the source distribution $\frac{P_t(\boldsymbol{x},a)}{P_s(\boldsymbol{x},a)}$ is upper bounded by a constant $C$ for $(\boldsymbol{x}, a)$ in the source distribution, the base distribution is chosen in the way that $\mathbb{E}_{P_t(\boldsymbol{x},a,r)}[(r - \mu_0)^2]$ over the target distribution is upper bounded by a constant $H$, and source distribution has $n$ i.i.d. data*

samples. *Assuming there are m percentage of samples in the target distribution that have a non-zero density ratio $\mathcal{W}(\boldsymbol{x},a) = \frac{P_s(\boldsymbol{x},a)}{P_t(\boldsymbol{x},a)}$, the expected squared error in the target data distribution for the robust regression estimator is upper bounded as follows with probability at least $1-\delta$:*

$$\mathbb{E}_{P_t(\boldsymbol{x},a,r)}\left[(r-\hat{\mu}(\boldsymbol{x},a))^2\right] \leq mC\left[\frac{1}{n}\sum_{i=1}^n \sigma^2(\boldsymbol{x}_i,a_i) + \eta_1 + 4M\hat{\mathfrak{R}}(\mathcal{F}) + 3M^2\sqrt{\frac{\log\frac{2}{\delta}}{2n}}\right] + (1-m)H,$$

*where $\mathcal{F}$ is the function class of $f$ with $\sup_{f,f'\in\mathcal{F},\boldsymbol{x},a}|f(\boldsymbol{x},a)-f'(\boldsymbol{x},a)|\leq M$ and a Rademacher complexity of $\hat{\mathfrak{R}}$, and $\eta_1$ upper bounds the gradient of optimization algorithm when it converges, as shown in Equation 31.*

**Bias Analysis for DM Methods.** Based on the analysis of the expected error as above, we obtain the following upper bound for the bias of the proposed direct method estimators, DM-PS/DM-GCS.

**Corollary 1.** *Under the same assumptions as Theorem 1, the bias of the robust regression direct method estimators over the target data distribution are upper bounded as follows with probability at least $1-\delta$:*

$$\mathbb{E}_{P_t(\boldsymbol{x},a,r)}\left[(r-\hat{\mu}(\boldsymbol{x},a))\right] \leq \left[mC\left[\frac{1}{n}\sum_{i=1}^n \sigma^2(\boldsymbol{x}_i,a_i) + \eta_1 + 4M\hat{\mathfrak{R}}(\mathcal{F}) + 3M^2\sqrt{\frac{\log\frac{2}{\delta}}{2n}}\right] + (1-m)H\right]^{1/2} := \epsilon. \quad (14)$$

**Bias Analysis for DR Methods.** We further analyze the bias of the proposed doubly robust estimators, DR-PS/DR-GCS. We have the following upper bound.

**Theorem 2.** *Under the same assumptions as Theorem 1, the bias of the robust regression doubly robust estimators over the target distribution are upper bounded as follows with probability at least $1-\delta$:*

$$\left|\mathbb{E}_{P_t(\boldsymbol{x},a,r)}\left[\hat{V}_{\text{DR}-\text{PS/DR}-\text{GCS}} - V^T\right]\right| \leq \frac{C\eta_1}{l} + 2CM\hat{\mathfrak{R}}(\mathcal{F}) + 3CM\sqrt{\frac{\log\frac{2}{\delta}}{2n}} + \epsilon. \quad (15)$$

The proofs of Theorem 1 and 2 can be found in Appendix D.

**Robustness under Large Shift.** The above bias analysis shows that the proposed methods remain robust under large distribution shifts. Our bias analysis results consist of two parts, the first term is based on data with a non-zero density ratio where we reweight the target distribution into source distribution and use standard Rademacher complexity. The second term refers to the bounded error of the base distribution for target data that do not share support with the source data. In this case, the $m$ value, which is the percentage of target data points well-supported by the source distribution, represents the magnitude of the distribution shift. When the shift becomes larger, more proportion of the data will have a density ratio close to 0, and $m$ becomes smaller, corresponding to cases where the prediction relies more heavily on the base predictor $f_0$. Note that traditional ERM methods can have an arbitrarily bad performance when there is a severe non-overlapping issue between the source and target data distribution, such as large part of the target distribution is uncovered by the source distribution. Reweighting-based methods also become ineffective as the importance weights will have a zero denominator in a subset of data points. Our methods can effectively alleviate the issue as the bias of the proposed methods, which is shown in Equation 14 and Equation 15, will lean towards utilizing the base predictor $f_0$ and have a bounded error as $m$ becomes closer to 0. This makes our method more favorable than other methods under large shift settings, as further illustrated in the experiments section. In particular, we demonstrate how our method performs compared with other methods under different scales of shifting in Section 6.2. The experimental results show that our method remains robust and outperforms other methods, especially when the shift becomes large, as shown in Figure 4. Besides, we also provide a consistency analysis result for the settings where the density ratio is bounded away from 0 in Appendix D.1.

# 6 Experiments

## 6.1 Experiment Settings

**Datasets.** In line with the experimental settings employed in previous studies (Dudik et al., 2014; Wang et al., 2017; Farajtabar et al., 2018; Su et al., 2019b;a), we conduct experiments[*] on 9 UCI datasets by

---

[*]The code for the experiments is available at https://github.com/guoyihonggyh/Distributionally-Robust-Policy-Evaluation-under-General-Covariate-Shift-in-Contextual-Bandits.

transforming the classification problems to the contextual bandits setting. Specifically, the context is set to be the input feature of the dataset, while the policy is a classification model and the action is the predicted class from the model. The reward is 1 if the predicted class matches the ground truth label and 0 otherwise. More details about how the dataset is created can be found in Appendix E.1, and the statistics of these datasets are summarized in Table 1 as follows.

Table 1: Dataset Statistics.

| DATASET | GLASS | ECOLI | VEHICLE | YEAST | PAGEBLOK | OPTDIGITS | SATIMAGE | PENDIGITS | LETTER |
|---|---|---|---|---|---|---|---|---|---|
| ACTIONS | 6 | 8 | 4 | 10 | 5 | 10 | 6 | 10 | 26 |
| FEATURES | 9 | 7 | 18 | 103 | 10 | 64 | 36 | 16 | 16 |
| INSTANCES | 214 | 336 | 846 | 1484 | 5473 | 5620 | 6435 | 10992 | 20000 |

**Logging Policies.** We conducted experiments using three distinct categories of logging policies, encompassing 10 different policies. Specifically, we employed the softened policy, which is in line with previous off-policy evaluation (OPE) work (Farajtabar et al., 2018; Su et al., 2019a), as well as Tweak-1 ($\rho$) policy and Dirichlet ($\gamma$) policy, which were inspired by prior research on label shift (Lipton et al., 2018) and offline contextual bandit optimization (Yang et al., 2023). **1)** Softened policy: Softened policy is defined as $\pi_{(\lambda,\zeta)}(a|\boldsymbol{x}) = \lambda + \zeta u$ if $a = \hat{\psi}(x)$, where $u \sim \text{Uniform}(-0.5, 0.5)$ and $\hat{\psi}$ is a deterministic policy trained by logistic regression. The remaining probability mass, $1 - (\lambda + \zeta u)$, is evenly distributed among classes $a \neq \hat{\psi}(x)$. **2)** Tweak-1 ($\rho$): One class accounts for $\rho$ probability, and the other classes evenly share the remaining $1 - \rho$. **3)** Dirichlet ($\gamma$): A fixed probability distribution generated by Dirichlet distribution parameterized by $\gamma$. Compared with the softened policy, the Tweak-1 and Dirichlet policies create more probability values close to 0 or 1 among the arms, resulting in a larger distribution shift between the logging policy and the target policy.

**Target Policies.** We evaluate our method across various combinations of logging policies and three distinct types of target policies. **1)** Softened target policy: We use the softened policy with parameters $(\lambda, \zeta) = (0.9, 0)$. **2)** Softened perfect policy: We further apply the softened policy techniques to modify a perfect policy, which consists of a 100% accuracy classification model. **3)** Softened diverse perfect policy: We set the target policy value for each class as a random permutation of the sequence $\left[\frac{1}{n}, \frac{2}{n}, \ldots, 1\right]$, where $n$ represents the number of classes. This can be viewed as a discrete version of uniform sampling from the interval $[0,1]$.

**Covariate Shift on the Context.** In our study, we examine two different types of context shifts within the data. We uniformly sample context for the target data, while manipulating the distribution of the context in the logging data in the following two ways to create covariate shift. **1)** Gaussian covariate shift: We first apply Principal Component Analysis (PCA) to reduce the dimension. We then define a Gaussian distribution on the first principal component as the new data distribution. **2)** Tweak-1 covariate shift ($\omega$): Following the logic of the Tweak-1 ($\rho$) logging policy, we assign a specific class of data a higher weight $\omega$ and all other classes with weight 1, and then do weighted sampling. Detailed information about the policies and data generation process can be found in Appendix E.1.

**Estimation of the Density Ratio.** We consider both the density ratio $\mathcal{W}$ known and unknown conditions in our experiments. For $\mathcal{W}$ unknown cases, we need to estimate them. Specifically, in policy shift settings, we estimate the logging policy $\beta$ with a logistic regression model, where we use the context $\boldsymbol{x}$ in the logging data as features and the corresponding action $a$ as labels. For general covariate shift case, we also need to estimate $\frac{P_s(\boldsymbol{x})}{P_t(\boldsymbol{x})}$. We leverage a discriminative classifier to obtain $\frac{P(\text{source}|\boldsymbol{x})}{P(\text{target}|\boldsymbol{x})}$ using both the logging and the target context. For instance, we train a logistic regression model to classify whether the context $\boldsymbol{x}$ comes from the logging data or the target data. Then, according to Bayes' theorem, we can obtain the $\frac{P_s(\boldsymbol{x})}{P_t(\boldsymbol{x})}$ by $\frac{P(\text{source}|\boldsymbol{x})}{P(\text{target}|\boldsymbol{x})} = \frac{P_s(\boldsymbol{x})P(\text{source})}{P(\boldsymbol{x})} / \frac{P_t(\boldsymbol{x})P(\text{target})}{P(\boldsymbol{x})} = \frac{P_s(\boldsymbol{x})}{P_t(\boldsymbol{x})}$ (when the source and target data has the same size, i.e. $P(\text{source}) = P(\text{target})$) (Bickel et al., 2007).

**Baselines.** For our experimental comparisons, we consider several baseline methods, including Direct Method (DM), Inverse Propensity Score (IPS), Doubly Robust (DR) (Robins & Rotnitzky, 1995; Bang & Robins, 2005; Dudik et al., 2011), SnIPS, Self-Normalized Doubly Robust (SnDR) (Swaminathan & Joachims, 2015b), More Robust Doubly Robust Estimator (MRDR) (Farajtabar et al., 2018), and Distributionally Robust Policy Evaluation (DRPE) (Si et al., 2020). We include a variant of our proposed methods DM (R), shown in Equation 23, as a baseline by setting the density ratio in the robust regression to 1. More

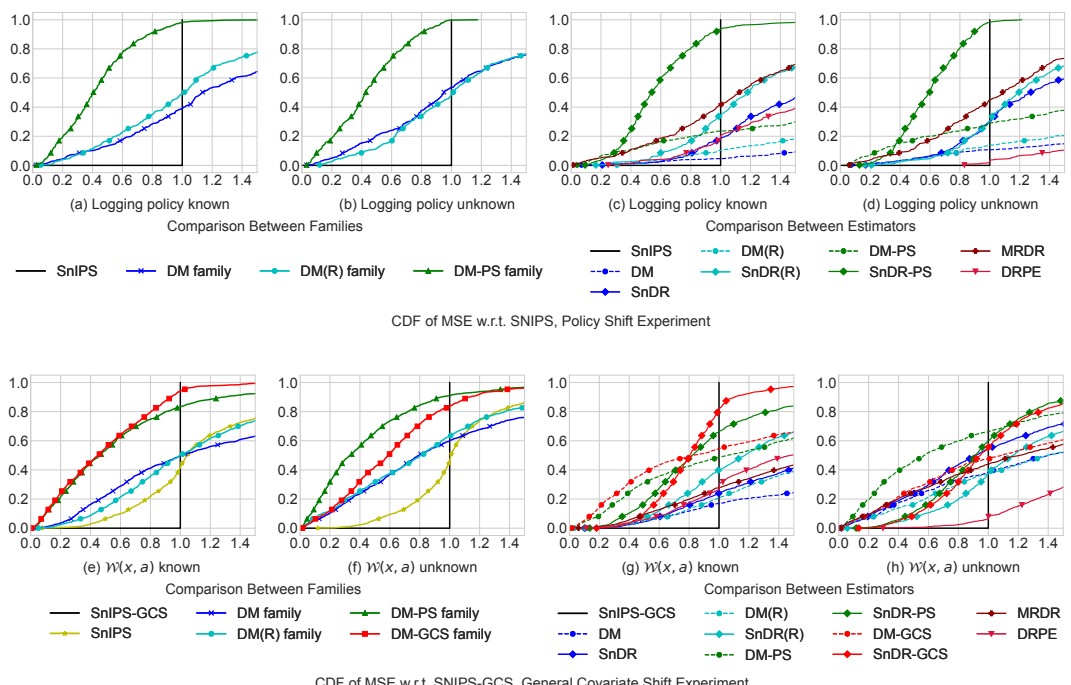

Figure 3: First row: Cumulative Distribution Function (CDF) of relative MSE w.r.t SnIPS when only policy shifts exist across 360 conditions. Second row: CDF of relative MSE w.r.t SnIPS-GCS when general covariate shifts exist across 1260 conditions. Subfigures (a)/(e) and (b)/(f) are comparisons between different families of methods, while (c)/(g) and (d)/(h) are comparisons between individual estimators. Better performance is indicated by CDF curves positioned in the top-left corner, signifying a relatively lower MSE. In the first row, among the evaluated methods, the DM-PS family is the best performer in known and unknown logging policy cases. In the second row, in the context of known $\mathcal{W}(x, a)$, the DM-GCS family performs best, while in scenarios with unknown $\mathcal{W}(x, a)$, the DM-PS family prevails.

details about the baselines, including SnIPS, SnDR, IPS-GCS, and DM(R), can be found in Appendix B. Additionally, we provide the further comparison with other OPE methods in Appendix E.4, including SWITCH (Wang et al., 2017) and Doubly robust estimator with Shrinkage (DRoS) (Su et al., 2020), demonstrating the advantage of our method when integrated with various kinds of OPE approaches.

To better compare our methods with baselines, we consider the corresponding families. Each DM estimator family comprises DM itself, DR, and SnDR employing the corresponding DM. Each family contains the same number of estimators for fair comparison. For instance, the DM-PS family includes DM-PS, DR-PS, and SnDR-PS. We select the best-performing method from each family for comparison in our experiments.

**Evaluation Metric.** We measure the performance of each estimator using Mean Squared Error (MSE), which is defined as $\mathrm{E}[(\hat{V} - V(\pi))^2]$, where $\hat{V}$ is the value calculated from the estimator and $V(\pi)$ is the ground truth value of the target policy on the target data distribution, both computed on a test set which consists of 25% of the dataset.

## 6.2 Experiment Results

We present the results for various combinations of logging policies, target policies, and covariate shifts on data, encompassing 360 conditions for the policy shift-only experiments and 1260 conditions for the general covariate shift experiments. Following the established setting in previous studies (Wang et al., 2017; Su et al., 2019a), we show the Cumulative Distribution Function (CDF) of relative MSE with respect to SnIPS for the policy shift and CDF of relative MSE with respect to SnIPS-GCS for the general covariate shift as the evaluation metric in Figure 3. Our analysis covers both cases where the logging policy is known (Subfigures (a), (c), (e), and (g)) and unknown (Subfigures (b), (d), (f) and (h)).

Table 2: The number of conditions where each family is the best under general covariate shift. DM-PS and DM-GCS families outperform baseline methods in over 72% of the conditions. When comparing the settings of $P_s(\boldsymbol{x}, a)$ known with $P_s(\boldsymbol{x}, a)$ unknown, the DM-GCS family outperforms DM-GCS family among known settings while the DM-PS family excels among unknown settings.

|  | SnIPS-GCS | DM | DM(R) | DM-PS | DM-GCS |
|---|---|---|---|---|---|
| $P_s(\boldsymbol{x}, a)$ known | 40 | 148 | 93 | 372 | 607 |
| $P_s(\boldsymbol{x}, a)$ unknown | 60 | 265 | 84 | 690 | 161 |

Table 3: The number of conditions where DM-PS or SnDR-PS is the best under different Tweak-1 logging policies. From left to right, the shift becomes larger. This suggests that SnDR-PS's advantage can be weakened by the worsened performance of the IPS component and disappear, as SnDR-PS consists of the DM-PS and the IPS component.

|  | Tweak-1 ($\rho = 0.91$) | Tweak-1 ($\rho = 0.95$) | Tweak-1 ($\rho = 0.99$) |
|---|---|---|---|
| DM-PS | 6 | 8 | 18 |
| SnDR-PS | 30 | 28 | 18 |

**General Performance Comparison:** From Figure 3 (a) and (b), which compare the performance across different method families under policy shift, we can see that the DM-PS family consistently outperforms other methods substantially. In Table 4 in Appendix E.2, we further present the number of conditions where our proposed method families perform optimally. The DM-PS family emerges as the top performer in both known and unknown logging policies, outperforming other baseline methods in over 90% of the conditions. For general covariate shift settings, when the density ratio $\mathcal{W}(\boldsymbol{x}, a)$ is known (Figure 3 (e)), the DM-GCS family exhibits the best performance. When $\mathcal{W}(\boldsymbol{x}, a)$ is unknown (Figure 3 (f)), the DM-PS family leads in performance. In both scenarios (Figure 3 (e) and (f)), both DM-GCS and DM-PS families outperform SnIPS-GCS in over 70% of the examined conditions. In (c), (d), (g), and (h), DM-PS and DM-GCS significantly outperform MRDR and DRPE. Further comparisons with advanced OPE methods such as SWITCH and DRoS are provided in Appendix E.4.

**Comparison between DM-PS and DM-GCS:** When comparing DM-PS and DM-GCS in Figure 3 (g) and (h) as well as in Table 2, we observe that DM-PS and its family perform better when the density ratio $\mathcal{W}(\boldsymbol{x}, a)$ is unknown, while DM-GCS and its family excels when $\mathcal{W}(\boldsymbol{x}, a)$ is known. These results can be attributed to that the GCS setting introduces the disparity between $P_s(\boldsymbol{x})$ and $P_t(\boldsymbol{x})$, creates larger distribution shifts and harder-to-estimate density ratios, and destabilizes the training of DM-GCS and the IPS component within DR methods. We further demonstrate this by comparing SnIPS and SnIPS-GCS in Table 5 in Appendix E.3. Also, the relatively weaker performance of DM-GCS when $\mathcal{W}(\boldsymbol{x}, a)$ is unknown can be attributed to the estimation of $P_s(\boldsymbol{x}, a)$, which requires a concurrent estimation of both $P_s(\boldsymbol{x})$ and $\beta(a|\boldsymbol{x})$. The inherent difficulty in accurately estimating $P_s(\boldsymbol{x})$, which is typically more complex than estimating the logging policy, negatively affects the performance of the DM-GCS family.

**Comparison between DM Methods and DR Methods:** In Figure 3 (c) and (d), by comparing the DM-PS with DM(R) and DM, we see that our method shows the best performance, highlighting the advantages of accommodating covariate shifts in reward estimation. We also observe that the SnDR-PS has the best performance. However, in the left corner, we see instances where the DM-PS outperforms the SnDR-PS, indicating that although SnDR-PS generally outperforms the DM-PS, as the shift becomes larger, its advantage is weakened by the worsened performance of IPS component. This is further demonstrated in Table 3, where we show the number of conditions where DM-PS and SnDR-PS are the best under different Tweak-1 logging policies. A larger $\rho$ represents a larger shift. One can see that when the shift becomes larger, the SnDR-PS's advantage becomes smaller. Furthermore, in the general covariate shift setting, we can still observe that DM-PS outperforms SnDR-PS in many conditions shown in Figure 3 (c) and (d), a pattern that is also seen between DM-GCS and SnDR-GCS. This can be explained by the negative effect on performance when incorporating the unreliable IPS into the SnDR method when the distribution shift is large. Additionally, when the distribution shift of $P_s(\boldsymbol{x})$ occurs, obtaining a reliable estimation of $\beta(a|\boldsymbol{x})$ becomes increasingly challenging, consequently affecting the estimation of the density ratio $\mathcal{W}(\boldsymbol{x}, a)$, and harming the performance of SnDR methods.

**The Effect of Different Shift Magnitudes:** We analyze the performance of DM-PS and DM-GCS under various scales of policy and covariate shifts to show how shifts affect the performance of the DM methods.

**1. Policy Shift.** Consider two types of softened shift logging policies: small shifts, such as $\pi_{(0.95,0)}(a|\boldsymbol{x})$ and $\pi_{(0.7,0.1)}(a|\boldsymbol{x})$, where logging policy deviations from the target policy are modest; and large shifts, represented by $\pi_{(0.5,0.1)}(a|\boldsymbol{x})$ and $\pi_{(0.1,0)}(a|\boldsymbol{x})$, that exhibit more substantial deviations. The Tweak-1 and Dirichlet policies are treated as even larger shift logging policies due to more probability values close to 0 or 1. Figure 4 (a) illustrates the proportion of each DM method achieving the best performance across these conditions, revealing that DM-PS consistently surpasses other methods across diverse logging policy scenarios. DM-PS's advantage grows as the shift enlarges, demonstrating its excellence in accommodating policy shifts in reward estimation and aligning well with our theoretical results in Section 5.1.1.

**2. General Covariate Shift.** Figure 4 (b) showcases the CDF of the relative MSE of DM-GCS with respect to DM-PS in Gaussian covariate shift scenarios, with a larger $\omega$ indicating a larger covariate shift. As the covariate shift increases, DM-GCS exhibits widening advantages over DM-PS, indicating the benefits of accommodating covariate shifts in reward estimation.

Collectively, these analyses highlight the superior performance of DM-PS and DM-GCS in handling varying scales of covariate shifts in reward estimation. They consistently exceed other methods, particularly as shifts become more pronounced.

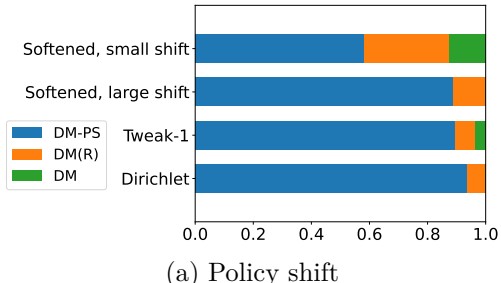
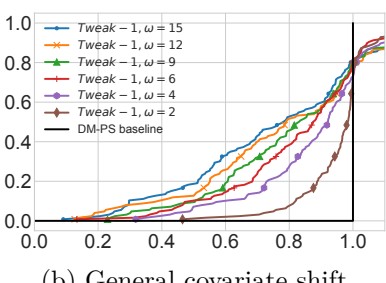

(a) Policy shift

(b) General covariate shift

Figure 4: **(a)** The proportion of each DM method achieving the best performance across conditions. As the covariate shift increases, the advantages of DM-PS become more pronounced, signifying the importance of considering covariate shifts in reward estimation. **(b)** The CDF of the relative MSE of DM-GCS with respect to DM-PS in Gaussian covariate shift cases. As the Gaussian covariate shifts on context increase, DM-GCS demonstrates more advantages.

## 7  CONCLUSION AND FUTURE WORK

We introduced a distributionally robust approach for policy evaluation under general covariate shifts in contextual bandits using robust regression. Utilizing this approach, we developed a novel family of policy evaluation methods for not only the policy shift setting but also the general covariate shift setting. We showed how the reward model from such distributionally robust reward estimators can be integrated into direct methods and doubly robust policy estimators. Theoretically, we established the finite sample upper bounds on the bias of the proposed methods, providing advantages in large shift situations. Empirically, we conducted extensive studies across a wide range of varying distribution shift conditions, demonstrating the superior robustness of our method. Future work includes the investigation of applying the distributional robustness idea to the OPE in reinforcement learning, offline, and safe reinforcement learning.

### Acknowledgement

This work is partially supported by the JHU-Amazon AI2AI faculty award, the Discovery Award of the Johns Hopkins University, and a seed grant from JHU Institute of Assured Autonomy.

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

# A    Related Work

**Off-Policy Evaluation for Contextual Bandits.**  Classic approaches for OPE (which only handle the policy shift setting) can be summarized into three categories:  direct methods (DM), inverse propensity scoring (IPS), and doubly robust (DR) methods that combine DM and IPS. DMs aim to directly learn the mapping from context and action to reward, but they can suffer from significant bias (Dudik et al., 2011) due to the divergence between the target and logging policies. IPS tackles the problem using the idea of important weighting between policy distribution. While unbiased, they exhibit high variance, especially under large policy shifts, due to unstable estimations of the density ratio (Dudik et al., 2011). A more recently proposed approach is the self-normalized IPS estimator (SnIPS) (Swaminathan & Joachims, 2015a), which offers improved accuracy over standard IPS when fluctuations in weights dominate that of the rewards.

DR methods (Dudik et al., 2011; 2014; Wang et al., 2017; Farajtabar et al., 2018) strike a balance between the biased, low-variance DM and the unbiased, high-variance IPS by incorporating both. It can also be interpreted as using control variates within an IPS framework (Thomas & Brunskill, 2016; Veness et al., 2011). If either IPS or DM is unbiased, DR methods are guaranteed to be unbiased (Dudik et al., 2011). Much of the follow-up work on DR methods has focused on mitigating the detrimental effects of variance or extreme probabilities from the IPS component. For example, SnDR utilizes SnIPS (Swaminathan & Joachims, 2015b), SWITCH estimator truncates IPS using a threshold (Wang et al., 2017), and DR with Optimistic Shrinkage (DRoS) finds a weighting strategy by optimizing a sharp bound on mean squared error (Su et al., 2019a).  While much of the prior work has emphasized improving the IPS component of DR estimators and their asymptotical guarantees, our research targets enhancing the robustness of conditional reward estimators, especially under large shifts and limited data. Our key insight is to view this problem as a covariate shift problem and utilize robust regression to train the conditional reward estimator.  Our proposed method can also be naturally incorporated into existing methods like DMs and DRs by offering a distributionally robust reward estimator.

**Covariate Shift and Distributional Robustness.**  Covariate shift is a specific type of distribution shift, and has been extensively studied in the machine learning community (Shimodaira, 2000).  Various methods have been developed to mitigate this challenge, including importance weighting (Shimodaira, 2000; Sugiyama et al., 2007) and kernel mean matching (Gretton et al., 2009). In the context of policy evaluation, Uehara et al. (2020) introduced a DR-based estimator under covariate shift that uses a density ratio estimator between the source and target distributions. However, their method focuses on the propensity score weighting formulation, and the IPS component in the estimator introduces high variance when extreme weights exist. In contrast, we tackle the problem orthogonally in this paper, utilizing a distributional robust direct method that can be effortlessly combined with their method.

Distributionally robust optimization (DRO) (Ben-Tal et al., 2009; Słowik & Bottou, 2022), a prevalent approach to handling distribution shifts, introduces the idea of optimizing with respect to the most adverse scenarios over a defined family of distributions. This has led to different variants such as Wasserstein DRO (Blanchet et al., 2021).  A few studies have explored distributional robustness in policy evaluation. Si et al. (2020) and Kallus et al. (2022) augmented evaluation robustness by conservatively estimating reward mapping with KL-divergence uncertainty sets. However, in these methods, the uncertainty set covers all possible shifts in the joint distribution of context, action, and reward, which does not characterize the covariate shift specifically. As a result, the estimator can be overly robust and less effective. Another line of work utilize DRO for estimating confidence interval in the off-policy setting (Karampatziakis et al., 2020; Dai et al., 2020; Faury et al., 2020), following a generalized empirical likelihood approach (Duchi et al., 2021).  The differences from our approach lie in the formulation of the DRO problem.  Specifically, they do not use the conditional distribution of the target variable (the reward in our case) as the adversarial player, and the uncertainty set constraint is formulated by the perturbation of empirical distribution of the data with distance measured using f-divergence, rather than using feature matching as in our case. Robust regression (Chen et al., 2016; Liu & Ziebart, 2017) tailored distributional robustness methods specifically for the setting of covariate shift. It constructs a predictor that approximately matches training data statistics but is otherwise the most uncertain on the testing distribution by minimizing the worst-case expected target loss and obtaining a parametric form of the predicted output labels' probability distributions. We are the

first to employ robust regression methods for policy evaluation, enabling the refinement of direct methods and the further enhancement of doubly robust estimators.

**Causal Inference.** The task explored in this paper has close ties to causal inference (Athey, 2015; Joachims et al., 2021). A core challenge in evaluating the individual treatment effect (ITE) and the average treatment effect (ATE) in causal inference is the evaluation of a counterfactual policy. IPS and DR have been investigated for ATE estimation using linear models (Tan, 2006; Kang et al., 2007) and recent advancements have leveraged methods from domain adaptation and deep representation learning (Johansson et al., 2016). In particular, ITE estimation is challenged by the specific type of distribution shift where the context distribution changes between treatment and control groups (Shalit et al., 2017). There has also been work on using causal models to enhance off-policy evaluation results (Oberst & Sontag, 2019; Namkoong et al., 2020). Besides, the setting of covariate shift has been studied in the task of randomized controlled trials (RCT) in causal inference as an issue of lacking external validity (Pearl & Bareinboim, 2022). However, these works mainly focus on devising new identification strategies of causal estimands. In this work, we focus on developing robust learning methods for counterfactual reasoning from the perspective of covariate shift.

## B   Description of More Baseline Methods

**DM**. Given reward predictor $\hat{r}$, direct method estimate $\hat{V}_{\text{DM}}$ as:

$$\hat{V}_{\text{DM}} = \frac{1}{|S|} \sum_{\boldsymbol{x} \in S} \mathbb{E}_{a \sim \pi} \left[ \hat{r}(\boldsymbol{x}, a) \right]. \tag{16}$$

**IPS**. Inverse propensity scoring (IPS) uses important weighting on the entries in $S$ to reflect the relative probabilities of choosing some action $a$ by the target policy $\pi$ versus the logging policy $\beta$:

$$\hat{V}_{\text{IPS}} = \frac{1}{|S|} \sum_{(\boldsymbol{x}, a, r) \in S} \frac{\pi(a|\boldsymbol{x})}{\hat{\beta}(a|\boldsymbol{x})} r, \tag{17}$$

where $\hat{\beta}$ is the logging policy (estimated if $\beta$ is unknown).

**DR**. The basic formulation of the doubly robust estimator is:

$$\hat{V}_{\text{DR}} = \hat{V}_{\text{DM}} + \frac{1}{|S|} \sum_{(\boldsymbol{x}, a, r) \in S} \left[ \frac{(r - \hat{r}(\boldsymbol{x}, a))\pi(a|\boldsymbol{x})}{\hat{\beta}(a|\boldsymbol{x})} \right]. \tag{18}$$

**SnIPS**. The self-normalized IPS (Swaminathan & Joachims, 2015b) is normalized by the sum of the importance weights. It follows the form as below:

$$\hat{V}_{\text{SnIPS}} = \frac{\sum_{(\boldsymbol{x}, a, r) \in S} r \frac{\pi(a|\boldsymbol{x})}{\hat{\beta}(a|\boldsymbol{x})}}{\sum_{(\boldsymbol{x}, a, r) \in S} \frac{\pi(a|\boldsymbol{x})}{\hat{\beta}(a|\boldsymbol{x})}}. \tag{19}$$

**SnDR**. The self-normalized doubly robust estimator normalizes the importance weights following the idea of SnIPS. The formulation of the SnDR is:

$$\hat{V}_{\text{SnDR}} = \frac{\sum_{(\boldsymbol{x}, a, r) \in S} \frac{(r - \hat{r}(\boldsymbol{x}, a))\pi(a|\boldsymbol{x})}{\hat{\beta}(a|\boldsymbol{x})}}{\sum_{(\boldsymbol{x}, a, r) \in S} \frac{\pi(a|\boldsymbol{x})}{\hat{\beta}(a|\boldsymbol{x})}} + \hat{V}_{\text{DM}}. \tag{20}$$

**IPS-GCS**. When general covariate shift exists, the density ratio of IPS is $\frac{P_t(\boldsymbol{x}, a)}{P_s(\boldsymbol{x}, a)}$. And the "IPS under general covariate shift" (IPS-GCS) is calculated by:

$$\hat{V}_{\text{IPS-GCS}} = \frac{1}{|S|} \sum_{(\boldsymbol{x}, a, r) \in S} r \frac{P_t(\boldsymbol{x}, a)}{P_s(\boldsymbol{x}, a)}, \tag{21}$$

where $P_t(\boldsymbol{x}, a) = P_t(\boldsymbol{x})\pi(a|\boldsymbol{x})$ and $P_s(\boldsymbol{x}, a) = P_s(\boldsymbol{x})\hat{\beta}(a|\boldsymbol{x})$.

**DM (R)**. We further obtain the baseline method DM (R) by setting $\mathcal{W}(\boldsymbol{x}, a) = 1$ in the robust regression framework. Such a framework does not consider covariate shift information. The solution of DM (R) is formulated as follows:

$$
\begin{aligned}
\mu_{\boldsymbol{\theta}}^{(\mathrm{R})}(\boldsymbol{x}, a) &= \sigma^2(\boldsymbol{x}, a)\left(-2\boldsymbol{\theta}_{\boldsymbol{x}}\boldsymbol{\phi}(\boldsymbol{x}, a) + \mu_0\sigma_0^{-2}\right), \\
\sigma_{\boldsymbol{\theta}}^{2(R)}(\boldsymbol{x}, a) &= \left(2\theta_r + \sigma_0^{-2}\right)^{-1}.
\end{aligned}
\tag{22}
$$

We then plug the mean estimates from Equation 22 into the direct method and obtain:

$$
\hat{V}_{\mathrm{DM\ (R)}} = \frac{1}{|S|}\sum_{\boldsymbol{x}\in S}\mathbb{E}_{a\sim\pi}\left[\mu_{\boldsymbol{\theta}}^{(\mathrm{R})}(\boldsymbol{x}, a)\right].
\tag{23}
$$

## C    Derivation of Robust Regression Model

We here show the derivation of the robust regression model. According to Chen et al. (2016), if we have the constraints for the adversary $g$ as follows,

$$
\Sigma_S := \left\{g \left| \left\| \mathbb{E}_{\boldsymbol{x}, a\sim P_s(\boldsymbol{x}, a), r\sim g}[\boldsymbol{\delta}(r, \boldsymbol{x}, a)] - \frac{1}{|S|}\sum_{i\in S}\boldsymbol{\delta}(r, \boldsymbol{x}, a) \right\| \leq \eta \right.\right\},
\tag{24}
$$

then the solution of the minimax formulation has the parametric form:

$$
\begin{aligned}
\hat{f}_{\boldsymbol{\theta}}(r|\boldsymbol{x}, a) &\propto f_0(r|\boldsymbol{x}, a)\exp\left(-\frac{P_s(\boldsymbol{x})\beta(a|\boldsymbol{x})}{P_t(\boldsymbol{x})\pi(a|\boldsymbol{x})}\boldsymbol{\theta}^T\boldsymbol{\delta}(r, \boldsymbol{x}, a)\right) \\
&= f_0(r|\boldsymbol{x}, a)\exp\left(-\frac{P_s(\boldsymbol{x}, a)}{P_t(\boldsymbol{x}, a)}\boldsymbol{\theta}^T\boldsymbol{\delta}(r, \boldsymbol{x}, a)\right).
\end{aligned}
\tag{25}
$$

If we set the function $\boldsymbol{\delta}$ in a specific way such that it has a quadratic form and assuming the base distribution $f_0 \sim \mathcal{N}(\mu_0, \sigma_0)$, we obtain a Gaussian distribution. We provide the necessary details here for the derivation of the mean and variance from this special form and refer more details to Chen et al. (2016).

If our $\boldsymbol{\delta} := \text{vector}\left([r, \boldsymbol{\phi}(\boldsymbol{x}, a)]^T[r, \boldsymbol{\phi}(\boldsymbol{x}, a)]\right)$, it corresponds to the following constraints in $\Sigma$:

$$
\begin{aligned}
\Sigma_S := \bigg\{g \bigg| & \left\| \mathbb{E}_{\boldsymbol{x}, a\sim P_s(\boldsymbol{x}, a), r\sim g}[r(\boldsymbol{x}, a)^2] - \frac{1}{|S|}\sum_{i\in S}r_i^2 \right\| \leq \eta, \\
& \left\| \mathbb{E}_{\boldsymbol{x}\sim P_s, a\sim\beta, r\sim g}[r(\boldsymbol{x}, a)\boldsymbol{\phi}(\boldsymbol{x}, a)] - \frac{1}{|S|}\sum_{i\in S}r_i\boldsymbol{\phi}(\boldsymbol{x}, a) \right\| \leq \eta\bigg\},
\end{aligned}
\tag{26}
$$

With strong duality and the constraint set for the adversary $g$ in Equation 24, the solution of loss function Equation 3 is equivalent to the solution of the following minimization problem:

$$
\begin{aligned}
&\min_{\hat{f}} D_{\boldsymbol{x}, a\sim P_t(\boldsymbol{x}, a), r\sim\hat{f}}(\hat{f}\|f_0), \\
&\text{such that: } \mathbb{E}_{\boldsymbol{x}, a\sim P_s(\boldsymbol{x}, a), r\sim\hat{f}}[\delta(r, \boldsymbol{x}, a)] = c,
\end{aligned}
$$

where $c = \frac{1}{n}\sum_i \delta(r_i, \boldsymbol{x}_i, a_i)$. Let $\hat{f}_{\boldsymbol{\theta}}$ be defined as:

$$
\hat{f}_{\boldsymbol{\theta}} = \frac{f_0(r|\boldsymbol{x}, a)\exp\left(-\mathcal{W}(\boldsymbol{x}, a)\boldsymbol{\theta}^T\delta(r, \boldsymbol{x}, a)\right)}{Z(\boldsymbol{x}, a, \boldsymbol{\theta})},
$$

where $Z(\boldsymbol{x}, a, \theta)$ denote the normalization term. Moreover, let

$$\Phi(r, \boldsymbol{x}, a) = -\mathcal{W}(\boldsymbol{x}, a)\boldsymbol{\theta}^T \delta(r, \boldsymbol{x}, a) - \log Z(\boldsymbol{x}, a, \boldsymbol{\theta}),$$

and use the $\boldsymbol{\theta}$ as the Lagrangian parameters, the Lagrangian of the optimization problem is written as:

$$\mathcal{L}(\boldsymbol{\theta}) = D_{\boldsymbol{x}, a \sim P_t(\boldsymbol{x}, a), r \sim \hat{f}}(\hat{f} \| f_0) + \boldsymbol{\theta}^T \left( \mathbb{E}_{P_s(\boldsymbol{x}, a, r)}[\delta(r, \boldsymbol{x}, a)] - c \right)$$

$$= \int_{\boldsymbol{x}, a} P_t(\boldsymbol{x})\pi(a|\boldsymbol{x}) \int_{r|\boldsymbol{x}, a} \hat{f}(r|\boldsymbol{x}, a) \log(\frac{\hat{f}}{f_0}) + \boldsymbol{\theta}^T \left( \mathbb{E}_{P_s(\boldsymbol{x}, a, r)}[\delta(r, \boldsymbol{x}, a)] - c \right)$$

$$= \int_{\boldsymbol{x}, a} P_t(\boldsymbol{x})\pi(a|\boldsymbol{x}) \int_{r|\boldsymbol{x}, a} \hat{f}(r|\boldsymbol{x}, a)(-\mathcal{W}(\boldsymbol{x}, a)\boldsymbol{\theta}^T \delta(r, \boldsymbol{x}, a)) - \int_{\boldsymbol{x}, a} P_t(\boldsymbol{x})\pi(a|\boldsymbol{x}) \int_{r|\boldsymbol{x}, a} \hat{f}(r|\boldsymbol{x}, a) \log Z(\boldsymbol{x}, a, \boldsymbol{\theta})$$

$$+ \boldsymbol{\theta}^T \left( \int_{\boldsymbol{x}, a} P_s(\boldsymbol{x})\beta(a|\boldsymbol{x}) \int_{r|\boldsymbol{x}, a} \hat{f}(r|\boldsymbol{x}, a)\delta(r, \boldsymbol{x}, a) - c \right)$$

$$= - \int_{\boldsymbol{x}, a} P_t(\boldsymbol{x})\pi(a|\boldsymbol{x}) \log Z(\boldsymbol{x}, a, \boldsymbol{\theta}) - \boldsymbol{\theta}^T c.$$

If we take the maximum of $\mathcal{L}(\boldsymbol{\theta})$, we have the following result:

$$\arg\max_{\boldsymbol{\theta}} \mathcal{L} = \arg\max_{\boldsymbol{\theta}} - \int_{\boldsymbol{x}, a} P_t(\boldsymbol{x})\pi(a|\boldsymbol{x}) \log Z(\boldsymbol{x}, a, \boldsymbol{\theta}) - \boldsymbol{\theta}^T c$$

$$= \arg\max_{\boldsymbol{\theta}} - \int_{\boldsymbol{x}, a} P_t(\boldsymbol{x})\pi(a|\boldsymbol{x}) \log Z(\boldsymbol{x}, a, \boldsymbol{\theta}) - \boldsymbol{\theta}^T \int_{\boldsymbol{x}, a} P_s(\boldsymbol{x})\beta(a|\boldsymbol{x}) \int_{r|\boldsymbol{x}, a} P^*(r|\boldsymbol{x}, a)\delta(r, \boldsymbol{x}, a)$$

$$= \arg\max_{\boldsymbol{\theta}} - \int_{\boldsymbol{x}, a} P_t(\boldsymbol{x})\pi(a|\boldsymbol{x}) \log Z(\boldsymbol{x}, a, \boldsymbol{\theta}) - \boldsymbol{\theta}^T \int_{\boldsymbol{x}, a} P_t(\boldsymbol{x})\pi(a|\boldsymbol{x})\mathcal{W}(\boldsymbol{x}, a) \int_{r|\boldsymbol{x}, a} P^*(r|\boldsymbol{x}, a)\delta(r, \boldsymbol{x}, a)$$

$$= \arg\max_{\boldsymbol{\theta}} \int_{\boldsymbol{x}, a} P_t(\boldsymbol{x})\pi(a|\boldsymbol{x}) \int_{r|\boldsymbol{x}, a} P^*(r|\boldsymbol{x}, a) \left( \log \frac{1}{Z(\boldsymbol{x}, a, \boldsymbol{\theta})} - \mathcal{W}(\boldsymbol{x}, a)\boldsymbol{\theta}^T \delta(r, \boldsymbol{x}, a) \right)$$

$$= \arg\max_{\boldsymbol{\theta}} \mathbb{E}_{\boldsymbol{x}, a \sim P_t(\boldsymbol{x}, a), r \sim P^*} \left[ \log \hat{f}_{\boldsymbol{\theta}}(\boldsymbol{x}, a) \right].$$

If we use parameters $\boldsymbol{\theta} := [\theta_r, \boldsymbol{\theta_x}, \boldsymbol{\theta_x}, \boldsymbol{\theta_{xx}}]$ as the Lagrangian parameters (here $\boldsymbol{\theta_x}$ is a vector with the same dimension as $\boldsymbol{\phi}$ ), we can derive the following predictive form for $\hat{f}_{\boldsymbol{\theta}}(r|\boldsymbol{x}, a)$ (note that $\boldsymbol{\theta_{xx}}$ is only relevant to $\boldsymbol{\phi}(\boldsymbol{x}, a)$ but not $r$, so we do not need to learn it explicitly):

$$\hat{f}_{\boldsymbol{\theta}}(r|\boldsymbol{x}, a) \propto f_0 \exp\left( -\frac{P_s(\boldsymbol{x}, a)}{P_t(\boldsymbol{x}, a)}\boldsymbol{\theta}^T \boldsymbol{\delta}(r, \boldsymbol{x}, a) \right) = f_0 \exp\left( -\frac{P_s(\boldsymbol{x}, a)}{P_t(\boldsymbol{x}, a)}(\theta_r r^2 + 2\boldsymbol{\theta_x} r\phi(\boldsymbol{x}, a) + \boldsymbol{\theta_{xx}}\phi(\boldsymbol{x}, a)^2) \right)$$

$$= \frac{1}{\sqrt{2\pi}\sigma_0} \exp\left( -\frac{1}{2}\frac{(r - \mu_0)^2}{\sigma_0^2} \right) \cdot \exp\left( -\frac{P_s(\boldsymbol{x}, a)}{P_t(\boldsymbol{x}, a)}(\theta_r r^2 + 2\boldsymbol{\theta_x} r\phi(\boldsymbol{x}, a) + \boldsymbol{\theta_{xx}}\phi(\boldsymbol{x}, a)^2) \right) \quad (27)$$

$$\propto \exp\left( -\left( \frac{1}{2\sigma_0^2} + \frac{P_s(\boldsymbol{x}, a)}{P_t(\boldsymbol{x}, a)}\theta_r \right) r^2 + (-2\frac{P_s(\boldsymbol{x}, a)}{P_t(\boldsymbol{x}, a)}\boldsymbol{\theta_x}\phi(\boldsymbol{x}, a) + \frac{\mu_0}{\sigma_0^2})r \right) \propto \exp\left( -\frac{1}{2}\frac{(r - \mu_{\boldsymbol{\theta}})^2}{\sigma_{\boldsymbol{\theta}}^2} \right), \quad (28)$$

where

$$\sigma_{\boldsymbol{\theta}}^2(\boldsymbol{x}, a) = \left( 2\frac{P_s(\boldsymbol{x}, a)}{P_t(\boldsymbol{x}, a)}\theta_r + \sigma_0^{-2} \right)^{-1}, \quad (29)$$

$$\mu_{\boldsymbol{\theta}}(\boldsymbol{x}, a) = \sigma_{\boldsymbol{\theta}}^2(\boldsymbol{x}, a) \left( -2\frac{P_s(\boldsymbol{x}, a)}{P_t(\boldsymbol{x}, a)}\boldsymbol{\theta_x}\phi(\boldsymbol{x}, a) + \mu_0\sigma_0^{-2} \right). \quad (30)$$

## C.1 Gaussian Predictive Form

Here we provide an example that the data is generated from cubic form distribution instead of Gaussian distribution. In this scenario, we can adjust the robust regression model by incorporating the cubic function

feature constraint instead of the quadratic constraint. Denote $x$ as the covariate variable, $y$ as the target variable, $f_0(y|x)$ as the prior and $\phi(x, y)$ to be the cubic function feature constraint. We can obtain the derived predictive model form of $p(y|x) \propto f_0(y|x) \exp(-w(x)\theta\phi(x, y))$, where $\theta$ is the model parameter and $w(x)$ is the density ratio. This estimator has a hard-to-track normalization denominator for the predictive model distribution, so we need to sample from this distribution to obtain the prediction of $y$ from such a model, which will sacrifice the nice implementation convenience of our robust regression model, and we may need to tailor specific sampling methods to efficiently obtain the prediction from such an estimator.

We conduct experiments under settings where $p(y|x)$ is not Gaussian by generating synthetic data $(x, y)$ such that $p(y|x) \propto \exp(y^3 - 2y^2x - 3yx^2 - 4y^2 + 5yx + 6y)$ following a cubic feature constraint assumption. Specifically, we created the distribution shift by generating $x$ following $x_{\text{train}} \sim N(0, 1)$ and $x_{\text{test}} \sim N(1, 2)$. There are 500 data points in both the train and test datasets. Since the cubic feature constraint does not result in a Gaussian distribution predictor model like the quadratic feature setting and the denominator of such distribution is intractable, we use discrete target $y$ by sampling $y \in \{0,1\}$ from the aforementioned prediction distribution in this experiment, for both the data generation and getting predictions from the robust regression model.

The cubic form robust regression achieved a mean squared error (MSE) of $0.31(\pm 0.00029)$. The quadratic form robust regression received an MSE of $0.42(\pm 0.0003)$, and the linear regression model received an MSE of $0.49(\pm 1.2 \times 10^{-10})$. These results are averaged over 10 runs. These results suggest that a more tailored feature set could improve the performance in synthetic settings and using a simpler hypothesis class could induce bias. However, given the original model for regression without Gaussian assumption, $p(y|x) \propto \exp(-w(x)\theta\phi(x, y))$, can be less convenient for making predictions (which is the reason we need to use sampling to get the $y$ in this synthetic experiments) and the broad applicability of Gaussian data assumptions, a robust regression model with Gaussian assumptions is still preferred in practice.

## D   Proof of Bias Analysis

**Expected Error.** We prove the bias analysis results of the proposed robust regression estimators under large distributional shift, building upon established results from Liu et al. (2019). We assume the selection of a base distribution $\mathcal{N}(\mu_0, \sigma_0^2)$, such that the expected squared error over the target distribution $\mathbb{E}_{P_t(\boldsymbol{x}, a, r)}[(r - \mu_0)^2]$ is upper bounded by some constant $H$. Typically, this condition is achieved in practice by appropriately setting the base distribution, such as to be the empirical mean of the dataset.

Our analysis begins with establishing a generalization bound for the expected squared error of the proposed robust regression-based direct method over the target distribution. We divide the target distribution data into two disjoint subsets based on whether it shares support with the source distribution, which is indicated by the value of its density ratio between the source and target distribution. Specifically, for the target distribution data points $(\boldsymbol{x}, a)$ with $\mathcal{W}(\boldsymbol{x}, a) \to 0$, which means they do not share support with the source distribution, the robust regression estimator is reduced to the base estimator based on the formulation. Consequently, under the aforementioned assumption, the following inequality holds:

$$\mathbb{E}_{P_t(\boldsymbol{x}, a, r)}\left[(r - \mu_0)^2\right] \leq H.$$

For the remaining data points in the target distribution with $\mathcal{W}(\boldsymbol{x}, a) > 0$, we assume that $\frac{P_t(\boldsymbol{x}, a)}{P_s(\boldsymbol{x}, a)}$, the density ratio of target over the source distribution, is upper bounded by some constant $C$. Drawing upon Theorem 1 from Liu et al. (2019), we derive the following upper bound for the expected squared error over the target distribution:

$$\mathbb{E}_{P_t(\boldsymbol{x}, a, r)}[(r - \hat{\mu}(\boldsymbol{x}, a))^2] \leq C\left[\frac{1}{n}\sum_{i=1}^{n}|r^2 - \hat{\mu}(\boldsymbol{x}, a)^2| + 4M\hat{\mathfrak{R}}(\mathcal{F}) + 3M^2\sqrt{\frac{\log\frac{2}{\delta}}{2n}}\right],$$

where $\mathcal{F}$ is the function class of $f$ with $\sup_{f,f'\in\mathcal{F},\boldsymbol{x},a}|f(\boldsymbol{x},a)-f'(\boldsymbol{x},a)|\leq M$, and with a Rademacher complexity of $\hat{\mathfrak{R}}(\mathcal{F})$. Assuming the training process converges satisfactorily, we can upper bound the training gradient and derive the following results:

$$
\begin{aligned}
r^2 - (\mu(\boldsymbol{x},a)^2 + \sigma^2(\boldsymbol{x},a)) &\leq \eta_1; \\
r - \mu(\boldsymbol{x},a) &\leq \frac{\eta_1}{l},
\end{aligned}
\tag{31}
$$

where $\eta_1$ is some small constant and $l$ is the lower bound of all the features in the context and action pair representation $f(x,a)$. Without loss of generality, we assume that there are $n$ i.i.d sampled data samples in training and $m$ percentage of samples have a non-zero density ratio $\mathcal{W}(x,a)$ in the target data distribution. We have the following upper bound for the expected error on the target distribution holds with probability at least $1-\delta$:

$$
\mathbb{E}_{P_t(\boldsymbol{x},a,r)}[(r-\hat{\mu}(\boldsymbol{x},a))^2] \leq mC\left[\frac{1}{n}\sum_{i=1}^{n}\sigma^2(\boldsymbol{x}_i,a_i) + \eta_1 + 4M\hat{\mathfrak{R}}(\mathcal{F}) + 3M^2\sqrt{\frac{\log\frac{2}{\delta}}{2n}}\right] + (1-m)H.
$$

**Bias Analysis of DM.** Building upon our analysis of the expected squared error of the proposed direct method using robust regression as above, we establish an upper bound for the bias of the robust regression-based direct method estimator. The following bound for the bias of DM-PS/DM-GCS holds with probability at least $1-\delta$:

$$
\begin{aligned}
\mathbb{E}_{P_t(\boldsymbol{x},a,r)}[(r-\hat{\mu}(\boldsymbol{x},a))] &\leq \left[mC\left[\frac{1}{n}\sum_{i=1}^{n}\sigma^2(\boldsymbol{x}_i,a_i) + \eta_1 + 4M\hat{\mathfrak{R}}(\mathcal{F}) + 3M^2\sqrt{\frac{\log\frac{2}{\delta}}{2n}}\right] + (1-m)H\right]^{1/2} \\
&:= \epsilon.
\end{aligned}
$$

**Bias Analysis of DR.** We then analyze the bias of the proposed DR-PS and DR-GCS estimators. Similarly, assuming the training process converges satisfactorily, we have the following lemma by upper bounding the training gradient and using Rademacher Complexity.

**Lemma 1.** *The expectation of the $L_1$ distance between the reward and the mean estimator from robust regression over the source distribution satisfies the following with probability at least $1-\delta$:*

$$
\mathbb{E}_{P_s(\boldsymbol{x},a,r)}\left[|r-\mu(\boldsymbol{x},a)|\right] \leq \frac{\eta_1}{l} + 2M\hat{\mathfrak{R}}(\mathcal{F}) + 3M\sqrt{\frac{\log\frac{2}{\delta}}{2n}},
\tag{32}
$$

*where $l$ is the lower bound of all the features in the context and action pair representation $f(x,a)$.*

Given that Equation 32 in Lemma 1 holds, we establish the following bound for the bias of DR-PS/DR-GCS:

$$
\left|\mathbb{E}_{P_t(\boldsymbol{x},a,r)}[\hat{V}-V_\pi]\right| \leq \left|\mathbb{E}_{P_s(\boldsymbol{x},a,r)}\left[C(r-\hat{\mu}(\boldsymbol{x},a))\right]\right| + \left|\mathbb{E}_{P_t(\boldsymbol{x},a,r)}\left[r-\hat{\mu}(\boldsymbol{x},a)\right]\right|
\tag{33}
$$

$$
\leq \frac{C\eta_1}{l} + 2CM\hat{\mathfrak{R}}(\mathcal{F}) + 3CM\sqrt{\frac{\log\frac{2}{\delta}}{2n}} + \epsilon
\tag{34}
$$

### D.1 Consistency

We here analyze the asymptotic convergence of the proposed DM-PS/DM-GCS and DR-PS/DR-GCS estimators using the asymptotic performance of robust regression method (Chen et al., 2016; Fathony et al., 2016). We have the following conclusions.

**Proposition 1.** *Assuming $\mathbb{E}_{P_t(\boldsymbol{x},a)}[\mathcal{W}(\boldsymbol{x},a)]$ is bounded away from zero, $\mathcal{W}(\boldsymbol{x},a)$ is accurately estimated in training, $\phi$ is chosen to be a universal kernel, then DM-PS/DM-GCS is consistent.*

**Remarks.** This proposition can be proved by directly applying the consistency results of the distributionally robust learning (Fathony et al., 2016). Here, the estimator will converge to zero error over the target distribution under the distribution shift. Note that the result is applicable for all loss functions.

We then analyze the asymptotic behavior of DR-PS/DR-GCS estimators.

**Proposition 2.** *If DM-PS/DM-GCS is consistent, then DR-PS/DR-GCS is consistent.*

**Remarks.** This proposition is obvious since we formulate the DR-PS/DR-GCS method as doubly robust methods with robust regression-based reward models. It demonstrates that using robust regression as a reward model does not hurt the asymptotic consistency of the doubly robust methods. Despite we need more assumptions for DM-PS/DM-GCS to be consistent, such as accurate $\mathcal{W}(x, a)$ and expressive feature representation $\boldsymbol{\phi}$, it does not mean that DR-PS/DR-GCS is weaker than general DR since we use a specific distributionally robust model for the reward model, while traditional DR analysis treats the reward model as a black box.

## E   Experimental details

We here describe the logging policy, target policy, and covariate we mentioned in Section 6.1 in detail and the specific parameters we used. We also provide experiment results on each dataset. The code for the experiments can be found at https://github.com/guoyihonggyh/Distributionally-Robust-Policy-Evaluation-under-General-Co–variate-Shift-in-Contextual-Bandits

### E.1   Data Generation

**Logging Policies.** We further describe our logging policy here. Firstly, we employed the softened policy, which is in line with previous work of OPE (Farajtabar et al., 2018; Su et al., 2019a). The softened policy is defined by two parameters, denoted as $(\lambda, \zeta)$, and a trained deterministic policy using logistic regression. Additionally, inspired by previous research on label shift (Lipton et al., 2018) and offline contextual bandit optimization (Yang et al., 2023), we generated the other two types of logging policies, Tweak-1 ($\rho$) and Dirichlet ($\gamma$), by creating biased "action label distributions".

- Softened policy, $\pi_{(\lambda,\zeta)}(a|\boldsymbol{x})$. Given a deterministic policy $\hat{\psi}$ and parameters $(\lambda, \zeta)$, the softened policy is defined as $\pi_{(\lambda,\zeta)}(a|\boldsymbol{x}) = \lambda + \zeta u$ if $a = \hat{\psi}(\boldsymbol{x})$ and other classes evenly share the $1 - (\lambda + \zeta u)$, where $u \sim \text{Uniform}(-0.5, 0.5)$. The deterministic policy $\hat{\psi}(x)$ is trained by logistic regression on 10% of training data. In our paper, we consider four different softened policies, which are $\pi_{(0.95,0)}(a|\boldsymbol{x})$, $\pi_{(0.7,0.1)}(a|\boldsymbol{x})$, $\pi_{(0.5,0.1)}(a|\boldsymbol{x})$ and $\pi_{(0.1,0)}(a|\boldsymbol{x})$.
- Tweak-1 ($\rho$). We allocate a probability of $\rho$ to one class while the remaining classes equally divide the residual probability of $1 - \rho$. A higher $\rho$ indicates a larger distribution shift of the logging dataset since more probability values close to 0 or 1 exist when $\rho$ is large. We evaluate this setting with $\rho$ values of $0.91, 0.95, 0.99$.
- Dirichlet ($\gamma$). Dirichlet ($\gamma$) represents a Dirichlet probability distribution parameterized by $\gamma$. A smaller $\gamma$ implies a larger shift of the logging dataset since there are more probability values close to 0 or 1. We generate one logging policy randomly and fix that across all experiments. Our study considers $\gamma$ values of $1.0, 0.5$, and $0.1$. Note that with Dirichlet (0.1), the probability for some classes becomes so low that they may not be sampled. Hence, for Dirichlet (0.1), we utilize a logging policy of $0.95 \times \text{Dirichlet}(0.1) + 0.05 \times \text{Uniform}$.

**Target Policies.** We test our method on various combinations of logging policies and target policies. The target policies are as follows:

- Softened policy, $\pi_{(0.9,0)}(a|\boldsymbol{x})$. We first obtain a deterministic policy $\psi$ by training a logistic regression on the entire training set. Then we set $(\lambda = 0.9, \zeta = 0)$ and obtain the softened policy.
- Softened perfect policy, $\pi_{(\text{perfect},\lambda)}$. Given access to the ground label when generating the policy, we can have the *perfect policy*, $\pi_{\text{perfect}}$, which means that for every $x$, the policy always chooses the action

that receives the highest reward. Following the idea of softened policy, we can soften the $\pi_{\text{perfect}}$ with $(\lambda, \zeta) = (\lambda, 0)$ to obtain $\pi_{(\text{perfect}, \lambda)}$. Our paper considers $\lambda = 0.9, 0.7, 0.5$, respectively.

- Diverse softened perfect policy. We create a case where the performance of policy evaluation will be substantially affected by the extent of the general covariate shift. As the covariate shift of the data we consider would generally make the target data only focus on a subset of all the classes in a dataset, we make the target policy value vary across different classes. Specifically, we set the target policy value for each class as a random permutation of $[\frac{1}{n}, \frac{2}{n}, ..., 1]$, where $n$ is the number of arms. This can be viewed as a discrete version of uniform sampling from [0,1].

**Covariate Shift on the Data.** We consider various context shifts in the experiments. We uniformly sample context for the target data while we consider the following choices for the scale of the distribution shift of the logging data distribution.

- Gaussian covariate shift. We first perform PCA on the dataset and select the first component $c$. We then define a Gaussian distribution with mean as $\min(c) + \frac{\min(c) - \text{mean}(c)}{a}$ and standard deviation as $\frac{\text{std}(c)}{b}$, where $a \in (0, \inf)$ and $b \in (0, 1)$. We then sample logging data based on the generated Gaussian distribution. Smaller $b$ means a smaller variance in the generated Gaussian distribution and a larger shift. Also, when $a > 1$, larger $a$ means a larger shift, when $a < 1$, smaller $a$ means a small shift. In our experiments, we consider the settings of $(a, b) = (1.5, 3), (2, 2), (1.5, 2), (0.6, 2)$, respectively.
- Tweak-1 covariate $(\omega)$. Following the idea of the $Tweak - 1(\rho)$ logging policy, we assign one class of data with a higher weight $\omega$ and other classes with weight 1. We can obtain a probability distribution for each data by normalizing the weight to 1. A larger $\omega$ means a larger shift. Our paper considers the setting of $\omega = 15, 12, 9, 6, 4, 2$, respectively.

When only policy shifts exist, we consider $9 \times 10 \times 4 = 360$ conditions derived from 9 datasets, 10 logging policies, and 4 target policies. When general covariate shifts exist, we consider $9 \times 7 \times 10 \times 2 = 1260$ conditions derived from 9 datasets, 7 logging policies, 10 covariate shifts, and 2 target policies. Specifically, the 7 logging policy are $\pi_{(0.95, 0.1)}(a|\boldsymbol{x}), \pi_{(0.7, 0.1)}(a|\boldsymbol{x})$, Tweak-1 $(\rho = 0.99)$, Tweak-1 $(\rho = 0.95)$, Tweak-1 $(\rho = 0.91)$, Dirichlet $(1.0)$ and Dirichlet $(0.1)$. The two target policies are $\pi_{(0.9, 0)}(a|\boldsymbol{x})$ and softened perfect policy $\pi_{(\text{perfect}, 0.7)}(a|\boldsymbol{x})$.

**Dataset Generation.** Based on the aforementioned settings, the dataset generation and evaluation process is as follows:

1. **Data Split:** We randomly split the original dataset into 75% training set $\mathcal{D}_{\text{TR}}$ and 25% test set $\mathcal{D}_{\text{TS}}$.
2. **Context and Policy Distribution Generation:** We generate logging policy $\beta$ and target policy $\pi$ given the policy parameters. If under the general covariate shift (GCS) setting, we further generate a logging context distribution $P_s(\boldsymbol{x})$.
3. **Target Policy Value Calculation:** We compute the ground truth of the target policy $\pi$'s value $V^T$ on the entire test set $\mathcal{D}_{\text{TS}}$, by sample the action $a$ for each context $\boldsymbol{x}$ in $\mathcal{D}_{\text{TS}}$ and get the corresponding reward $r$.
4. **Logging Dataset Generation:** If under the GCS setting, we sample context $\boldsymbol{x}$ from the training set $\mathcal{D}_{\text{TR}}$ based on the logging context distribution $P_s(\boldsymbol{x})$. Otherwise, we sample the logging context $\boldsymbol{x}$ uniformly. We then sample the action $a$ for each context $\boldsymbol{x}$ using the logging policy $\beta$ and obtain the corresponding reward $r$ to create the logging dataset $\mathcal{D}_{\text{TR-Logging}}$.
5. **Model Training:** We train the reward predictors, including DM, DM(R), DM-PS, and DM-GCS, using $(\boldsymbol{x}, a, r)$ in the logging dataset $\mathcal{D}_{\text{TR-Logging}}$.
6. **Evaluation:** We sample another logging dataset $\mathcal{D}_{\text{TS-Logging}}$ following the Logging Dataset Generation process in step 4 from the test set $\mathcal{D}_{\text{TS}}$ for estimating the importance sampling part of the estimators. All methods' final estimated policy value $\hat{V}$ is calculated on $\mathcal{D}_{\text{TS-Logging}}$.

**Hyperparameters.** The base distribution for robust regression is a Gaussian distribution with mean $= 0.6$ and variance $= 1$. $\theta$ is updated with SGD. We tune the hyperparameters with a grid search on the learning

rate in [0.001,0.0005]. We also set the learning rate decay for the learning of $\theta$, where the learning rate is multiplied by $\frac{10}{10+\sqrt{i-1}}$ at i-th epoch. The batch size is searched in [8, 32, 64, 256].

## E.2 Comparison of DM-PS Family and Other Baselines under Policy Shift

|  | SnIPS | DM | DM(R) | DM-PS |
|---|---|---|---|---|
| $\beta(a\|\boldsymbol{x})$ known | 9 | 7 | 10 | 334 |
| $\beta(a\|\boldsymbol{x})$ unknown | 3 | 17 | 13 | 327 |

Table 4: The number of conditions where each family is the best under policy shift. DM-PS family outperforms baseline methods in over 90% of the conditions.

## E.3 Comparison of SnIPS and SnIPS-GCS under different covariate shift

We also analyze the impact of various types of covariate shifts on the performance of SnIPS and SnIPS-GCS in Table 5 as below. Specifically, SnIPS exhibits superior performance under the Gaussian covariate shift, which generally creates a larger covariate shift compared to the Tweak-1 covariate shift, outperforming SnIPS-GCS across an extensive range of conditions. While under the Tweak-1 shift, SnIPS-GCS demonstrates enhanced performance in a larger number of scenarios than SnIPS. This observation demonstrates that large covariate shifts create harder-to-estimate density ratios and negatively affect the performance of SnIPS-GCS, compared to SnIPS.

Table 5: The number of conditions where each estimator is the best when $P_s(\boldsymbol{x}, a)$ is known.

| covariate shift | Target policy | SnIPS | SnIPS-GCS |
|---|---|---|---|
| Gaussian covariate | $\pi_{(0.9,0)}(a\|\boldsymbol{x})$ | 870 | 390 |
|  | Diverse Softened Perfect policy | 945 | 315 |
| Tweak-1 covariate | $\pi_{(0.9,0)}(a\|\boldsymbol{x})$ | 404 | 856 |
|  | Diverse Softened Perfect policy | 240 | 1020 |

## E.4 Further Comparison with Advanced OPE Methods

We present the result of our method applied to and compared with other advanced OPE approaches such as SWITCH (Wang et al., 2017) and DRoS (Su et al., 2020) in Figure 5. Our method can be integrated into SWITCH and DRoS by replacing their reward estimator with our estimators, DM-PS and DM-GCS. Incorporating DM-PS into SWITCH and DRoS results in SWITCH-PS and DRoS-PS, respectively. Similarly, applying DM-GCS to SWITCH and DRoS results in SWITCH-GCS and DRoS-GCS. As shown in Figure 5, SWITCH-PS and DRoS-PS consistently outperform the original SWITCH and DRoS methods, respectively, across all cases, including both policy shift and general covariate shift cases. Additionally, in the general covariate shift setting, the performance of DRoS is also further improved by DM-GCS. These results demonstrates the advantage of integrating DM-PS and DM-GCS into SWITCH and DRoS.

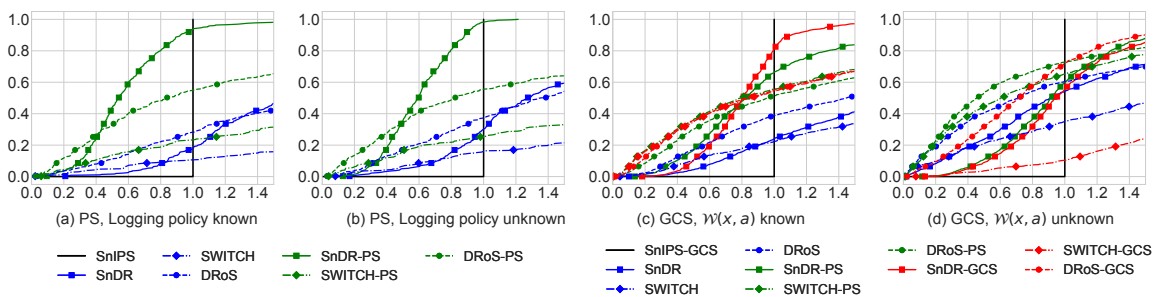

Figure 5: Subfigures (a) and (b) present the experiment results under policy shift. Subfigures (c) and (d) present the experiment results under general covariate shift. DM-PS and DM-GCS enhance the performance of SWITCH and DRoS.

### E.5 Experiments Results on Each Dataset

We show the experiment results on each dataset under policy shift and general covariate shift in Figure 6 and Figure 7 as below. Specifically, each row corresponds to the result of each dataset. In each row, subfigures (a) and (b) are comparisons between different families of methods, while (c) and (d) are comparisons between individual estimators. Subfigures (a) and (c) display the results when the $\beta(a|\boldsymbol{x})$ is known in policy shift conditions or $\mathcal{W}(x, a)$ is known in general covariate shift conditions, while (b) and (d) present the results when they are unknown.

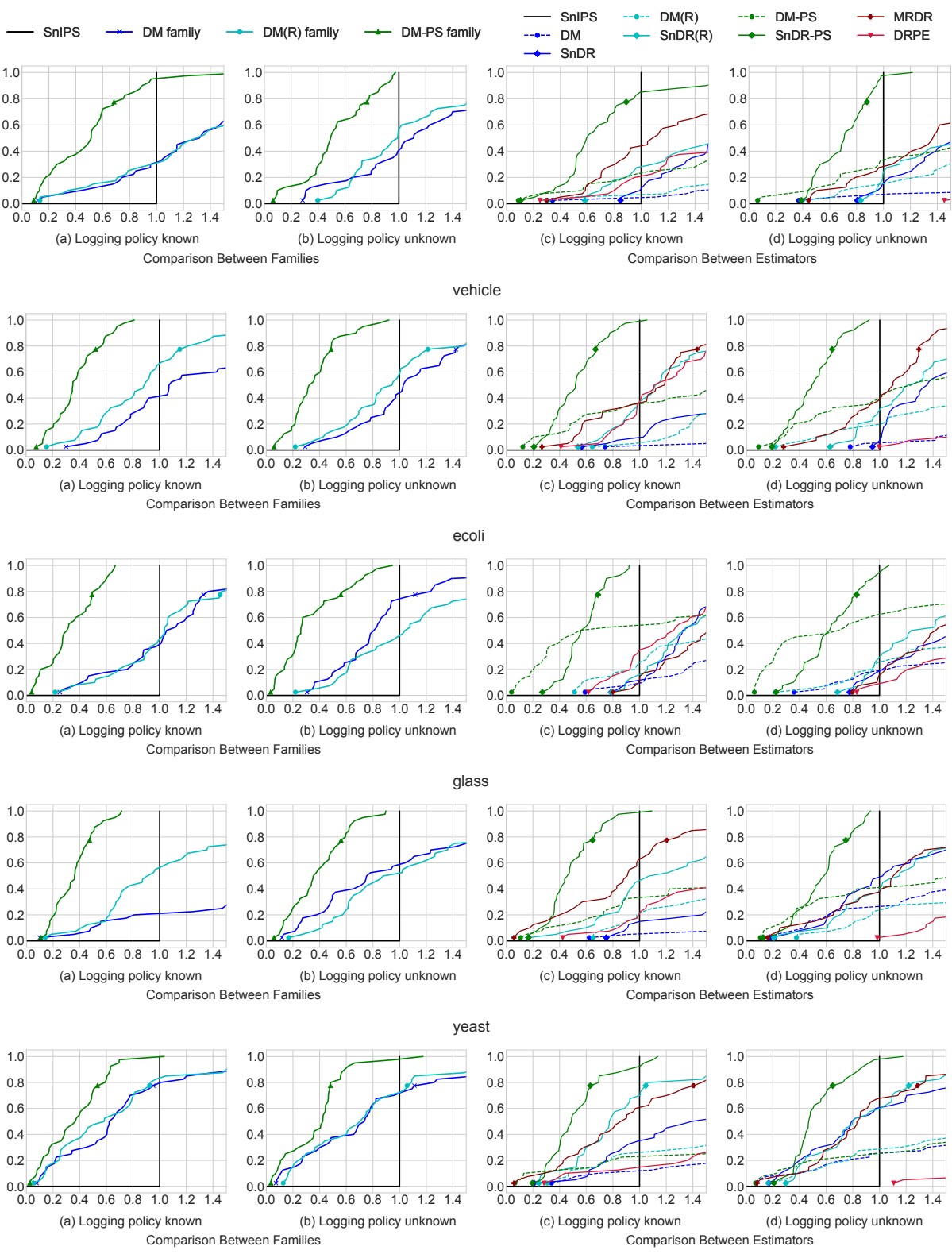

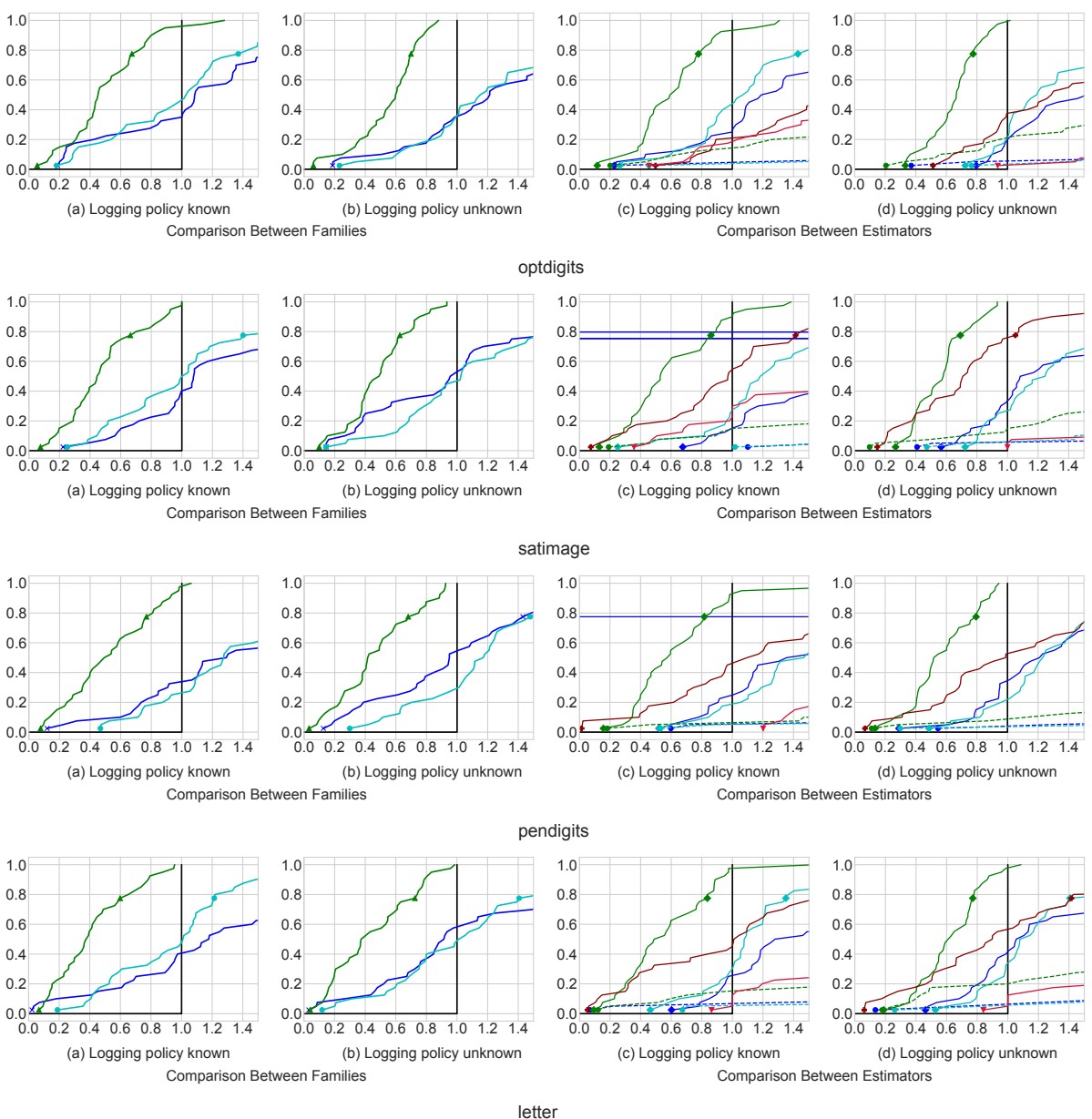

Figure 6: Experimental results of different datasets under policy shifts.

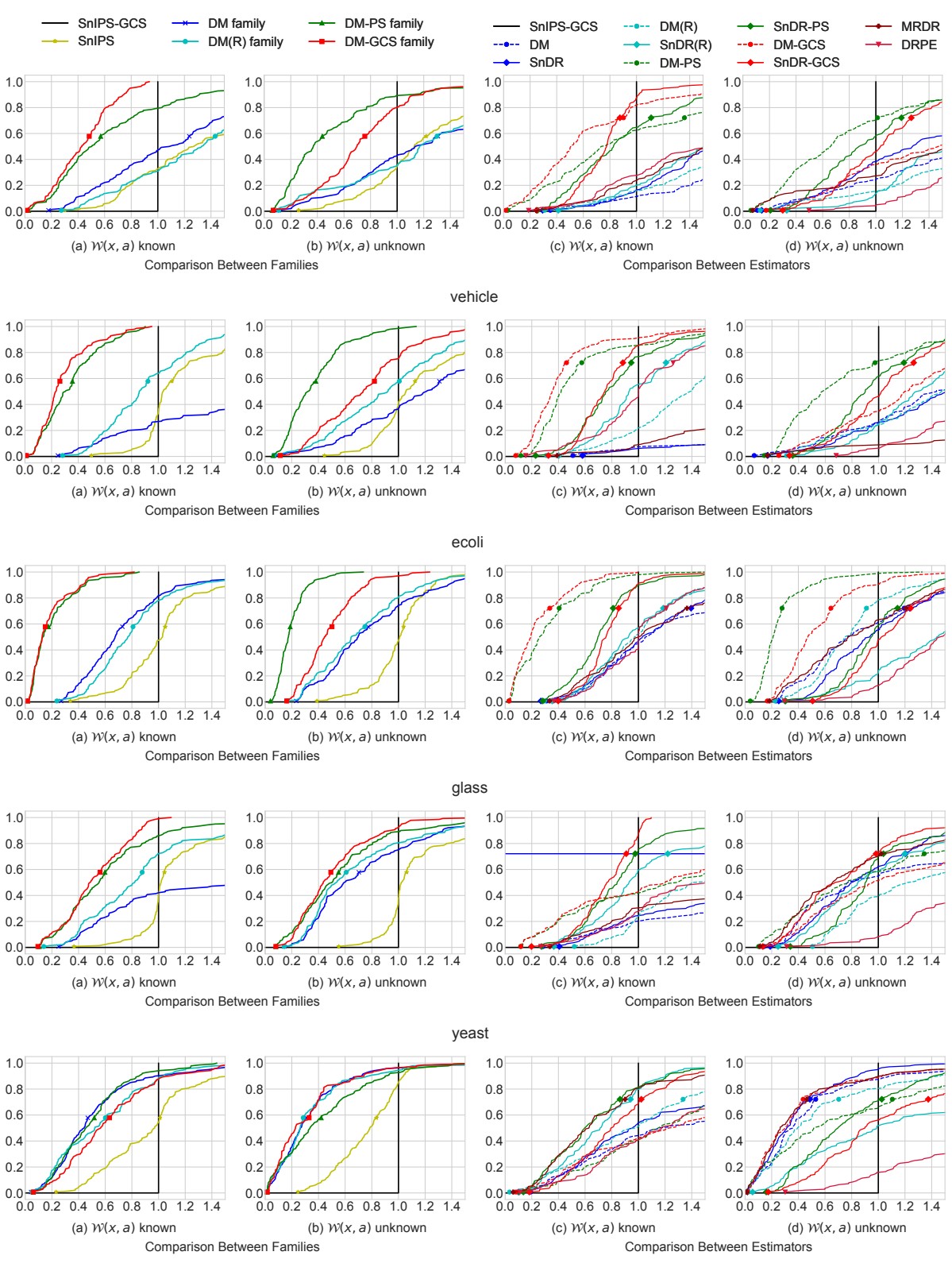

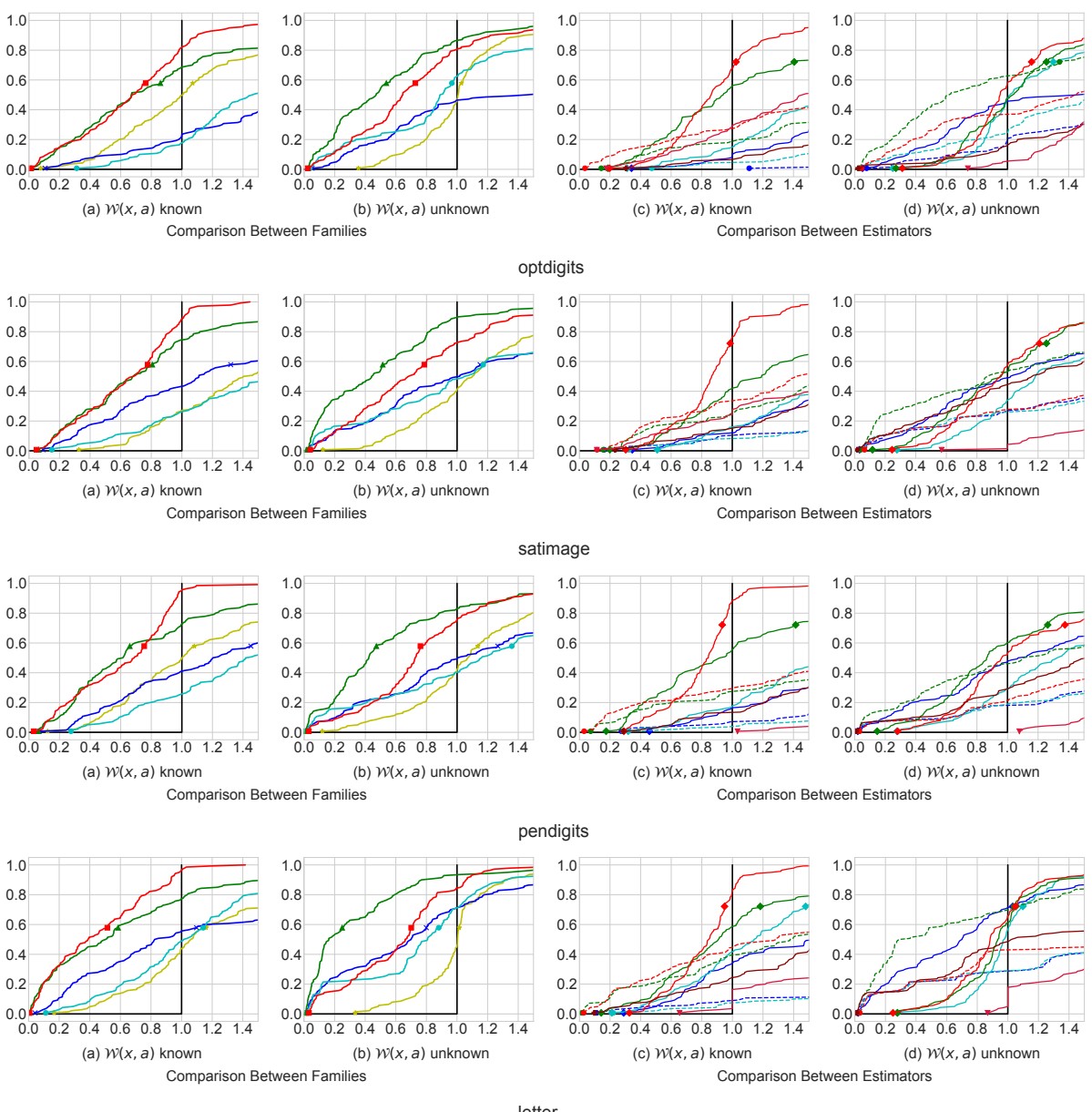

Figure 7: Experimental results of different datasets under general covariate shifts.

