# OpenReview forum: "Distributionally Robust Policy Evaluation under General Covariate Shift in Contextual Bandits"
_TMLR — Accepted by TMLR_

### Review · Reviewer_mVrZ · 2024-02-03

**Summary Of Contributions:**

This work considers the distribution shift in the contextual bandit setting, recovering the classic setting of off-policy evaluation. The authors interpret the policy evaluation as a bandit equivalent version of covariant shift, where the covariates are the contexts and actions and the dependent variable is the reward. Borrowing the ideas from distributional robust optimization, the authors propose a robust reward estimator and provide justifications about the error bounds. This work also conduct implausible number of combinations of covariate shift settings to evaluate their methods.

**Audience:**

Yes

**Broader Impact Concerns:**

While this paper does not present a Broader Impact Statement, I don't foresee any societal impacts that should be addressed in this context.

**Claims And Evidence:**

Yes

**Requested Changes:**

I recommend that the authors comprehensively present all the experimental results obtained for each combination of covariate shift settings.

**Strengths And Weaknesses:**

**Strengths**
In general, this paper did a good job in posing the research problem and articulating how they reached the solution. While the formulation of covariate shift in the contextual bandit setting may resemble that in the classification setting, I find the work to be original and novel.  The paper is clearly written and well organized. Notably, the authors implement the reward estimation methods into different modeling framework, and provide the corresponding error bound.

**Weaknesses**
- It is not intuitive to understand why the parameters $\theta$ are divided into $\theta_r$ and $\theta_x$ in Eq. (5). I would suggest the authors to further elaborate this in the *main* paper.
-  The authors claim that they conduct experiments under 1260 total combinations of datasets, logging policies, covariate shifts, and target policies. However, this claim appears somewhat exaggerated. Have the authors provided experimental results for all 1260 combinations, and does the performance consistently exhibit the same trend across this entire set of combinations?

---

> ### Author Response · Authors · 2024-02-16
>
> Dear reviewer, we would like to first thank you for your time and valuable comments. We address your concerns as follows:
>
> 1. **Division of $\theta$ into $\theta_r$ and $\theta_x$**: The reason for the partition of parameter $\theta$ is that the constraint of the adversary $g$ in the robust regression method is on the function $\delta$ which depends on both $x$ and $r$, hence in the derivation of the robust regression, the $\theta$ parameters are being divided into two components, $\theta_r$ is the weight for $r^2$, while $\theta_x$ is the weight for $r\phi$. This way of presentation is to highlight how the robust regression estimator, as well as the gradients for updating the parameters, depends on the different parameters. For instance, the variance of the estimator is only dependent on $\theta_r$ while the mean of the estimator depends on $\theta_x$. There are also different gradient update rules for $\theta_x$ and $\theta_r$, where $\theta_r$ update rule depends on $r^2$ while $\theta_x$ update rule depends on both $r$ and $\phi$.
>
>
> 2. **1260 combinations of data settings**: Figure 3 in our paper, which represents the CDF of relative MSE, indeed shows the trend of a large number of experiments of different data settings. This is because the figure measures the distribution of the value of different estimator’s performance w.r.t the SnIPS, which is obtained from a large number of different experiments, with each experiment setting only corresponding to one value for this variable. We also put the trend of the results of multi-experiments in each dataset in Appendix D.4. For more details, we provide the full experimental results in this link https://anonymous.4open.science/r/Result-of-Distributionally-Robust-Policy-Evaluation-under-General-Covariate-Shift-31D0/README.md
>
> Thank you once again for your valuable feedback. We hope that our clarifications adequately address your concerns.

---

### Review · Reviewer_ATLF · 2024-02-19

**Summary Of Contributions:**

This work studies methods to incorporate covariate shift into off policy evaluation, especially policy shift and context shift. Extensions of doubly robust estimators (which fuse together importance weighting and control variates) are developed for these settings, and their finite-sample analysis is provided. Experiments demonstrate the merits of the proposed methods relative to prior art especially in the context of covariate shift.

**Audience:**

Yes

**Broader Impact Concerns:**

See response to previous field.

**Claims And Evidence:**

Yes

**Requested Changes:**

See response to previous field.

**Strengths And Weaknesses:**

Strengths:

The technical setting studied, as well as the algorithms developed, are novel, to my knowledge.

Specifically, distributionally robust optimization approaches are developed for the off-policy evaluation objective. Their derived predictive forms is also novel, in my understanding, as well as stochastic gradient algorithms based upon them.

Experiments clearly demonstrate the merits of the proposed techniques for the setting of covariate shift.

 Weaknesses:

Convergence analysis seems to mostly follow from prior techniques, or otherwise it is not very clear what innovative steps are deployed here to establish the main theoretical results.

The class of distributional drifts that the method can address seem to require a specific model of distributional robustness that seems to be not specifically tethered to the dimensions in the covariate where the drift is largest, but instead defines a coarser notion of distributional robustness. It would be nice to see how a prior could be weighted to better target the robustness of the resultant estimator.

If one does not impose structural such that the derived predictor is Gaussian (aka non-quadratic losses), how much more predictive power does one gain? Are there situations where the Gaussian form is not valid? I would like to see some experimental studies with respect to classes of mismatch distributions, to better understand when this modeling assumption is/is not valid.

---

> ### Author Response · Authors · 2024-03-07
>
> Dear reviewer, we would like to first thank you for your time and valuable comments. We addressed your concerns as follows:
>
> 1.**Convergence Analysis:**
>
> Although our proof techniques are generally built upon existing techniques, the main goal of the analysis is to demonstrate how robust regression, which combines a prior base regressor and a data-driven learned regressor, will behave, particularly in situations under large shifts, which is effectively illustrated by our experimental results.
>
> 2.**Large distribution shift on certain dimensions:**
>
> Thank you for the excellent question. In our distributionally robust learning setup, the distribution shift is characterized by feature constraints. In other words, we aim to be robust against the worst-case data-generating distribution that satisfies a feature constraint. By allowing more errors or slacks in the specific dimensions, we can account for varying shift magnitudes in each feature dimension. Furthermore, imposing feature-specific constraints is straightforward in our framework.
>
> For example, if a certain covariate is drifting significantly, we can impose a larger slack term on that dimension. This can be interpreted within our algorithm framework by lifting the slack term $\eta$ in Equation (26) of the paper to be a vector and imposing different values on different dimensions of $\eta$, denoted as $\eta_i$. To tackle the slack term in the optimization, we can write out the dual formulation and analyze the KKT conditions. In this case, the full primal optimization problem of our method is:
>
> $\min(D(\hat{f}||f_0) + \xi \eta^2 $
>
> $s.t. E(\delta) - c - \eta <=0$.
>
> Its Lagrangian dual optimization problem can be written as:
>
> $\max  L = D(\hat{f}||f_0) + \xi \eta^2 + \theta (E(\delta) - c - \eta)$.
>
> Take the derivative of the dual $L$ w.r.t. $\eta$ to find the optimal condition of $\eta$, we obtain:
>
> $\eta = \theta / (2 \xi)$.
>
> Plugging this in, we can obtain the gradient for $\theta$, which will be $E(\delta) -c - \theta / \xi$.
>
> Setting $\lambda = 1 / \xi$, the gradient of $\theta$ become $E(\delta) - c - \lambda \theta$, which is equivalent to a dual problem with L2 dual regularization. Therefore, a feature-specific slack term corresponds to a feature-specific regularization weight. Here, a larger slack $\eta_i$ corresponds to a larger $\lambda_i$. This means we would like the regularization $\lambda_i$ to be larger and make the corresponding model parameters $\theta_i$ in that dimension closer to 0, such that the corresponding covariate provides less information for the prediction (compared to a smaller $\lambda_i$, such as $\lambda_i=0$) as the shifting on that covariate will be large. On the other hand, setting $\lambda_i = 0$ will correspond to an exact matching constraint.
>
> We conducted a synthetic experiment to demonstrate the effect of feature-specific slack terms. We created a synthetic dataset with covariates of two dimensions $(x_1, x_2)$. In the training set, we generated $x_1$ and $x_2$ following the Gaussian distribution, $x_1\sim N(0,10)$ and $x_2 \sim N(0,10)$. In the testing set, we generated $x_1 \sim N(20,10)$ and $x_2 \sim N(40,30)$. There are 500 data points in both the train and test datasets. The $x_2$ dimension has a larger distribution shift, compared to $x_1$, as the mean of $x_2$ in the testing set deviates more from the training data. We sample $y$ based on $y = I(x_1^2 + 3x_1 - 2x_2 +5 > 50)$, where $I$ is the indicator function.
>
> We show the visualization of the distribution of training and testing data in Figure 1 at (https://anonymous.4open.science/r/bandit_more_baseline-05FE/README.md) (covariate_data.png).  Note that the visualization figure has unequal axis aspect ratios for $x_1$ and $x_2$. Circle represents $y=1$, and “x” represents $y = 0$, while yellow denotes training data and red denotes testing data in Figure 1.
>
> We tested our algorithm with different slack terms and performed parameter tuning. The best slack parameters for this dataset are $(\lambda_1, \lambda_2)  = (0.1,0.5)$. The regression achieved a mean squared error (MSE) of $0.1648 (\pm 1.36 \times 10^{-5})$ without slack terms and $0.1554 (\pm 1.28 \times 10^{-5})$ with slack terms. The linear regression received an MSE of $0.41 (\pm 1.43 \times 10^{-4})$. These results are averaged over 10 runs. Our synthetic example demonstrates that we can tackle more specific feature shifts by leveraging the slack terms in the constraints in our method. We will clarify this and provide an ablation study in our revision.

---

> ### Author Response · Authors · 2024-03-07
>
> 3 **When Gaussian form is not valid:** Firstly, we would like to argue that the Gaussian assumption is relatively general as we essentially only require the conditional distribution $p(y|x)$ to be Gaussian, conditioning on a given $x$, where $x$ is the covariate and $y$ is the target. Similar assumptions have been widely applied in machine learning literature, such as variational autoencoder, etc. While a robust regression estimator is built from an exponential family, only when we impose the quadratic features constraint do we obtain the Gaussian estimator, which has various nice properties in implementation and practice.
>
> While this Gaussian assumption is generally valid for real-world examples, we can indeed build synthetic examples to violate such assumptions. For instance, we can make $p(y|x)$ an exponential family with cubic terms over the exponential, which also corresponds to a cubic function feature constraint. In this situation, the standard bias and variance tradeoff analysis applies –  if we apply the model with Gaussian assumption in this situation, we are essentially using an overly-simplified model, which will result in high bias and low variance.
>
> Furthermore, we can adjust the robust regression model for this situation by incorporating the cubic function feature constraint. Denote $f_0(y|x)$ as the prior and $\phi(x,y)$ to be the cubic function feature constraint, we can obtain the derived predictive model form of $p(y|x) \propto f_0(y|x) \exp(-w(x) \theta \phi(x,y))$. This estimator has a hard-to-track normalization denominator for the predictive model distribution. To obtain the prediction of $y$ from such a model, we need to sample from this distribution. This will sacrifice the nice implementation convenience of our robust regression model and we may need to tailor specific sampling methods to efficiently obtain the prediction from such an estimator.
>
> We conducted experiments under settings where $p(y|x)$ is not Gaussian by generating synthetic data $(x,y)$ such that $p(y|x)  \propto  \exp(y^3 - 2y^2x - 3yx^2 - 4y^2 + 5yx + 6y)$ following a cubic feature constraint assumption. Specifically, we created the distribution shift by generating $x$ following $x_{\text{train}} \sim N(0,1)$ and $x_{\text{test}} \sim N(1,2)$. There are 500 data points in both the train and test datasets.  Since the cubic feature constraint does not result in a Gaussian distribution predictor model like the quadratic feature setting and the denominator of such distribution is intractable, we use discrete target $y$ by sampling $y \in $ { 0,1 } from the aforementioned prediction distribution in this experiment, for both the data generation and getting predictions from the robust regression model.
>
> The cubic form robust regression achieved a mean squared error (MSE) of $0.31 (\pm 0.00029)$. The quadratic form robust regression received an MSE of $0.42 (\pm 0.0003)$, and the linear regression model received an MSE of $0.49 (\pm 1.2 \times 10^{-10} )$.  These results are averaged over 10 runs. These results suggest that a more tailored feature set could improve the performance in synthetic settings and using a simpler hypothesis class could induce bias. However, given the original model for regression without Gaussian assumption, $p(y|x) \propto \exp(-w(x) \theta \phi(x,y))$, can be less convenient for making predictions (which is the reason we need to use sampling to get the $y$ in this synthetic experiments) and the broad applicability of Gaussian data assumptions, a robust regression model with Gaussian assumptions is still preferred in practice. We will add a clarification in our final version.
>
> Thank you once again for your valuable feedback. We hope that our clarifications adequately address your concerns.

---

### Review · Reviewer_FLHu · 2024-02-22

**Summary Of Contributions:**

This paper studies OPE for contextual bandits, under general covariate shift conditions (both the covariate distribution and policy are changing). It mainly focus on how to improve the reward model part among the OPE components. Specifically, it views the OPE problem as covariate shift, and utilizes robust regression, a distributionally robust technique to improve the estimate of the reward distribution from logged data. The more advanced reward model (RM) estimator could be used as a plug-in estimator for both DM, DR, snDR, etc. Extensive semi-synthetic experiments conducted on contextual bandits showcase the method's efficacy compared to traditional DM approaches.

**Audience:**

Yes

**Claims And Evidence:**

Yes

**Requested Changes:**

See the Weakness Section.

**Strengths And Weaknesses:**

Strength:

- This paper is well-written and easy to read, it is interesting to look the OPE problem from the covariate shift perspective and the utilization of robust regression to improve DM in OPE is intuitive and well-justified.
- The method is simple to implement, and it introduces improvements without significantly increasing computational complexity over existing baselines.
- Meanwhile, the empirical results across various conditions (such as dataset, magnitude of policy/covariate shift, known/unknown density estimation, etc) is comprehensive, demonstrating the effectiveness of the advanced DM estimator, compared with the vanilla ones.

Weakness:

- The bias analysis for the regression estimator is interesting, but it does not provide much insight comparing only looking at the closed form solution in Equation (5). Theorem 2 on the upper bound of DR is confusing, ideally it should have a much simpler form by utilizing the "doubly robust" unbiased nature of the estimator, as here only need to bound the unsupported part?

- The justification of the constant prior is missing, is it for a simpler form of the estimator only? Ideally the prior could be based on any existing DM estimators and improve upon that.

- How is the optimality of the proposed robust regression estimator, say under a specific class of reward distributions?

- The empirical results seems missing some important baselines. Examples could include MRDR, variations of reweighting on the DM training loss function, localized doubly robust DROPE, and other general OPE estimators like SWITCH, DRoS, clipping, etc. This would provide a more comprehensive comparison and a clearer understanding of how the proposed reward estimator improvements compare with a broader range of existing methods. It would also be great to see how the reward estimators improvement further translate into the improvement of these advanced estimators.

Reference:
- More Robust Doubly Robust Off-policy Evaluation. Mehrdad Farajtabar, Yinlam Chow, Mohammad Ghavamzadeh
- Optimal and Adaptive Off-policy Evaluation in Contextual Bandits. Yu-Xiang Wang, Alekh Agarwal, Miroslav Dudik
- Doubly robust off-policy evaluation with shrinkage. Yi Su, Maria Dimakopoulou, Akshay Krishnamurthy, Miroslav Dudík
- Distributionally Robust Batch Contextual Bandits. Nian Si, Fan Zhang, Zhengyuan Zhou, Jose Blanchet
- Doubly Robust Distributionally Robust Off-Policy Evaluation and Learning. Nathan Kallus, Xiaojie Mao, Kaiwen Wang, Zhengyuan Zhou

---

> ### Author Response · Authors · 2024-03-07
>
> Dear reviewer, we would like to first thank you for your time and valuable comments. We address your concerns as follows:
>
> 1.**Simpler form of bias analysis:**
>
> To answer your question, we would like to first restate the main takeaways from our theoretical analysis. Our analysis, although being built on existing techniques, demonstrates how robust regression, a combination of a prior base regressor and a data-driven learned regressor, will behave and contribute to the policy evaluation problem, especially under large shift situations, which is effectively illustrated by our experimental results.
>
> Regarding the DR analysis, we would like to clarify that despite being unbiased asymptotically when either the IPS or DM method is unbiased, it does not necessarily mean the finite sample performance of DR would be better than DM. Asymptotically, DR “has two chances to succeed”, but in the finite sample regime, it may also have “two chances to fail”, especially when the shift is large or the sample is small. This is because DR now depends on two models and how the failure cases of them interacting with each other may depend on data. So our analysis focuses more on the finite sample guarantees and how the proposed method affects DR. We did not try to characterize the IPS part and how it may contribute to a better bias. Instead, we utilized an upper bound of the density ratio, which makes the bound looser but shows the “rather worse” cases of DR. We will add additional discussions about this in our final version.
>
> 2.**Existing DM estimators as prior:**
>
> Thank you for the great suggestion. First, we would like to argue that a fully data-driven prior model will not serve our interests. This is because, to achieve robustness,  the resulting predictive form of the robust regression framework essentially combines a predefined model that hardly relies on potentially skewed data and a data-driven component that relies on the data. If the prior model is also data-driven, the prior will again depend on potentially skewed data under shift, hence sacrificing the robustness of our framework.  However, in theory, we can indeed learn a specific class of reward model that generates different prior distributions for different inputs. We will leave this exploration for future work.
>
> 3.**Optimality of the robust regression estimator:**
>
> Our understanding is that the reviewer is asking about the conditional reward distribution given context and action. Our robust regression estimator is optimal if the data it applied to well fits the model assumption, i.e. quadratic feature constraint and conditional Gaussian target distribution. Similar to the third question of the reviewer ATLF under "When Gaussian form is not valid", while our model assumption is generally feasible, when there is a mismatch between the data distribution and the model assumptions, standard bias-variance tradeoffs can be applied.
>
> For example, if we apply the Gaussian distribution predictor, which corresponds to a quadratic feature, to the cubic feature constraint data generation setting, we are essentially using an overly simplified model, which will result in high bias and low variance.  Conversely, If we use an overly complicated model compared to the data generation setting,  it will result in low-bias but high-variance scenarios, as standard bias-variance tradeoff analysis suggests. We would like to kindly refer you to our responses to Reviewer ATLF for more details.

---

> ### Author Response · Authors · 2024-03-07
>
> 4.**More baselines:**
>
> Thank you for the references. Following the suggestion, we implemented additional baselines: Shrinkage [1], Switch [2], MRDR [3], and DRPE (distributionally robust policy evaluation) [4] on a subset of the datasets (Vehicle, Glass, and Optdigits) and logging/evaluation policy settings. We consider 36 conditions for policy shift experiments and 60 conditions for general covariate shift experiments due to limited time. We will complete all the experiments in our final version.
>
> We also agree with the reviewer that our method can be used to improve methods that depend on a plain direct method and do not account for shift and robustness. Therefore, we incorporate our model DM-PS/DM-GCS into the Shrinkage/Switch estimator to further improve model performance by obtaining “Switch with DM-PS” and “Shrinkage with DM-PS” for the policy shift setting, and “Switch with DM-GCS” and “Shrinkage with DM-GCS” for the general covariate shift setting. For MRDR and DRPE, we cannot incorporate our method. This is because MRDR trains a reward estimator with a weighted least square objective that minimizes the variance of the doubly robust estimator, which is difficult to be compatible with our parameter learning algorithm (shown in Section 4.2 of our paper).  DRPE is an important sampling-based method that doesn’t require a reward estimation direct method, while our method can be used to improve methods that depend on a not robust direct method. Also, another difference between DRPE and ours is that they consider the shift of the joint distribution of context, action, and reward and hence can be overly robust and less effective, while our method characterizes the covariate shift specifically.
>
> The results are shown in Figure 2 and Figure 3 at (https://anonymous.4open.science/r/bandit_more_baseline-05FE/README.md) (ps.png and gcs.png). Note that we do not show an estimator's performance if other methods significantly dominate its performance. We didn’t put the result from DROPE [5] as we faced severe numerical instability with their method in our implementations (NaN and overflow value).
>
> Our results show that in the policy shift-only setting, the SnDR-PS (one of our methods in the paper) and “Shrinkage with DM-PS” (using DM-PS estimator in shrinkage) have the best performance and significantly outperform baseline methods like MRDR, DRPE, Switch, and Shrinkage. Moreover, “Shrinkage with DM-PS” significantly improves the vanilla Shrinkage method. In the general covariate shift (GCS) setting, both DM-GCS and SnDR-GCS (Both are our methods in the paper) have the best performance and outperform other baselines. Additionally, the shrinkage and switch methods benefit from the DM-GCS estimator as  “Switch with DM-GCS / Shrinkage with DM-GCS” are both much better than “Switch with DM / Shrinkage with DM”.
>
> [1] Doubly robust off-policy evaluation with shrinkage. Yi Su, Maria Dimakopoulou, Akshay Krishnamurthy, Miroslav Dudík
>
> [2] Optimal and Adaptive Off-policy Evaluation in Contextual Bandits. Yu-Xiang Wang, Alekh Agarwal, Miroslav Dudik
>
> [3] More Robust Doubly Robust Off-policy Evaluation. Mehrdad Farajtabar, Yinlam Chow, Mohammad Ghavamzadeh
>
> [4] Distributionally Robust Batch Contextual Bandits. Nian Si, Fan Zhang, Zhengyuan Zhou, Jose Blanchet
>
> [5] Doubly Robust Distributionally Robust Off-Policy Evaluation and Learning. Nathan Kallus, Xiaojie Mao, Kaiwen Wang, Zhengyuan Zhou
>
> Thank you once again for your valuable feedback. We hope that our clarifications adequately address your concerns.

---

### Review · Reviewer_cCYM · 2024-02-23

**Summary Of Contributions:**

This work proposed a method to improve offline policy evaluation in contextual bandits. It discussed existing works and introduced Algorithm 1 where both source and target distributions are Gaussian distributions. I provided bias analysis for DM and DR methods and presented experimental results.

**Audience:**

Yes

**Claims And Evidence:**

Yes

**Requested Changes:**

Please refer to the 'Weaknesses' part in the 'Strengths and Weaknesses' section.

**Strengths And Weaknesses:**

Strengths:
1. The author(s) presented a detailed review of related works on offline policy evaluation.
1. The author(s) described how the difference between source and target distributions influence the density ratio $W(x,a)$ and hence the evaluation.

Weaknesses:
1. Algorithm 1 assumes that the contexts are drawn from Gaussian distribution. If this assumption does not hold, what should we expect theoretically and empirically? For example, what if source or target distribution is not Gaussian?

---

> ### Author Response · Authors · 2024-03-07
>
> Dear reviewer, we would like to first thank you for your time and valuable comments. We address your concerns as follows:
>
> **Assumption regarding the context distribution:**
>
> We would like to first clarify that we are not assuming the context is drawn from a Gaussian distribution. In general, we are not making any assumption about the context distribution $P(x)$. If the concern is instead, the resulting conditional reward distribution $P(r|x, a)$ being Gaussian, which is similar to the last question of Reviewer ATLF, we would like to kindly refer you to our responses to Reviewer ATLF under “When Gaussian form is not valid”.
>
> Briefly speaking, in theory and synthetic settings, we can benefit from a more specifically chosen model, other than a model under Gaussian assumption, but we will suffer from the inconvenience of implementation and making predictions, as demonstrated in our synthetic cubic feature constraint example in our responses to Reviewer ATLF. Theoretically, we may suffer from bias and inconsistency if there is a model mismatch. But in practice, when little is known about the underlying data generation process, the robust regression model under Gaussian assumption is preferred due to its nice properties of implementation and broad applicability. Please check our response to Reviewer ATLF under “When Gaussian form is not valid” for more details.
>
> Thank you once again for your valuable feedback. We hope that our clarifications adequately address your concerns.

---

### Decision · Action_Editor_ct2t · 2024-04-29

**Recommendation:** Accept with minor revision

**Comment:**

In this submission, the authors study OPE for contextual bandits under general covariate shift. Specifically, the authors exploit distributionally robust regression for reward estimation, which will be plugged for existing OPE estimator, e.g., DM, DR, and snDR.

The authors did comprehensive evaluation the proposed methods to justify the claim and provided theoretical analysis for the proposed method. Although the novelty of the proposed method is not significant, the TMLR does not include novelty and significancy as a criterion. However, there are several issues raised by the reviewers need to be address:

- The important baselines suggested by reviewer FLHu should be included in the final version.

-  The method derivation is based on Gaussian assumption (reviewer cCYM and ATLF). The limitation and potential extension beyond this Gaussian assumption should be included in the final version.

-  There are existing methods exploiting distributionally robust optimization for confidence interval of OPE (under the name "empirical likelihood") [1, 2, 3], which should be also discussed.

[1] Nikos Karampatziakis, John Langford, and Paul Mineiro. Empirical likelihood for contextual bandits. arXiv
preprint arXiv:1906.03323, 2019.

[2] Faury, Louis, Ugo Tanielian, Elvis Dohmatob, Elena Smirnova, and Flavian Vasile. "Distributionally robust counterfactual risk minimization." In Proceedings of the AAAI Conference on Artificial Intelligence, vol. 34, no. 04, pp. 3850-3857. 2020.

[3] Dai, Bo, Ofir Nachum, Yinlam Chow, Lihong Li, Csaba Szepesvári, and Dale Schuurmans. "Coindice: Off-policy confidence interval estimation." Advances in neural information processing systems 33 (2020): 9398-9411.

**Audience:**

The paper considers off-policy evaluation under general co-variate shift in contextual bandits, which is an important topic with wide audience in the community.

**Claims And Evidence:**

As all reviewers agreed, this paper is a solid work with strong empirical results for an important problem.